# Evaluating Long Range Dependency Handling in Code Generation LLMs

**Yannick Assogba**                                                     *yassogba@apple.com*
*Apple*

**Donghao Ren**                                                         *donghao@apple.com*
*Apple*

**Reviewed on OpenReview:** https://openreview.net/forum?id=yzACI2vFaX

## Abstract

As language models support larger and larger context sizes, evaluating their ability to make effective use of that context becomes increasingly important. We analyze the ability of several code generation models to handle long range dependencies using a suite of multi-step key retrieval tasks in context windows up to 8k tokens in length. The tasks progressively increase in difficulty and allow more nuanced evaluation of model capabilities than tests like the popular needle-in-the-haystack test. We find that performance degrades significantly for many models (up to 2x) when a function references another function that is defined later in the prompt. We also observe that models that use sliding window attention mechanisms have difficulty handling references further than the size of a single window. We perform simple prompt modifications using call graph information to improve multi-step retrieval performance up to 3x. Our analysis highlights ways that long-context performance needs deeper consideration beyond retrieval of single facts within a document.

## 1   Introduction

Long context inference is an increasingly important differentiator of LLMs, with model releases supporting larger and larger context windows (Anthropic, 2023; OpenAI, 2023; Google, 2024; Jiang et al., 2023; Touvron et al., 2023). This is enabled by advances in efficient attention implementation such as FlashAttention (Dao et al., 2022), Grouped-Query Attention (Ainslie et al., 2023) and Paged Attention (Kwon et al., 2023), as well as scaling attention by introducing sparsity via windowing (Child et al., 2019; Beltagy et al., 2020).

Applications, such as in-editor code completion from tools like GitHub's Copilot (Choi, 2024) and Source-Graph's Cody (Isken & Hill, 2024), benefit from long context support as they leverage retrieval-augmented generation strategies (Gao et al., 2024) to incorporate code snippets from across the user's project into a single prompt. This allows completions to be driven by the user's context rather than just by what the model learned at training time. Works such as Zhang et al. (2023) and Shrivastava et al. (2023) have proposed various approaches to improve cross-repository code completion.

These applications rely heavily on the model's ability to effectively use everything in its context window, and thus evaluation of long context usage has been a topic of interest in the community. Khandelwal et al. (2018) showed that early LSTM-based language models only roughly modelled information from more distant tokens. Recently, the "needle-in-the haystack" test, which measures recall for chat-style LLMs (Kamradt, 2024) has grown popular, and a parallel key-retrieval method has been used by Rozière et al. (2024) to measure recall in code generation models. However, we believe that desirable long context inference also includes the ability to reason over multiple pieces of information in the input, and needle-in-haystack evaluation approaches do not capture a model's ability to do this. In the paper we contribute:

1. A set of multi-step key retrieval tasks that progressively increase in difficulty and allow evaluation of long-range dependency handling and basic reasoning through function calls. [1]

2. Empirical evaluation of several open source and proprietary code generation models. We find that model performance varies greatly depending on the number of steps involved, and the distinctiveness of the target fact compared to the rest of the context. We also discover that the order of function declarations has a large effect on model ability to complete these tasks. We further observe that sliding window mechanisms degrade models' ability to resolve references beyond the size of the window.

3. An investigation of methods for improving multi-step recall that rely only on prompt modification using information obtainable via existing non-LLM techniques, in particular we add annotations of function dependencies in the form of comments.

## 2 Related Work

**Symbolic reasoning:** Zhang et al. (2022) study neural networks' ability to handle indirection by introducing the Pointer Value Retrieval (PVR) task. They train models to take in a sequence of tokens (typically digits), where the first token acts as pointer to the value to be retrieved. The synthetic nature of this task provides full control over the complexity of the problem. Abnar et al. (2023) extend this work to add recursive steps to the PVR task, evaluate it in the context of vision transformers, and present a new architecture that supports adaptive compute.

**Multi-hop QA:** Min et al. (2019) study a popular multi-hop reasoning benchmark (HotpotQA) and find that many of the questions can be answered with single hop reasoning. They find many questions embed contextual info that make the task easier, and that weak distractors may make picking the right answer 'obvious' without need for the desired reasoning steps. Chen & Durrett (2019) further explore dataset design choices to induce multi-hop reasoning.

**Long context retrieval:** Liu et al. (2024) examine instruction-tuned LLM performance for synthetic key-retrieval tasks as well as multi-document QA and describe a "lost-in-the-middle" phenomenon where information becomes harder to retrieve if it is not near the beginning or end of the prompt.

**Long context dependencies:** Yu et al. (2024) present a code completion benchmark named CoderEval, where problem solutions have varying levels of dependencies on code in the surrounding context. They find that model performance for functions with dependencies is significantly worse than performance for standalone functions. In the context of natural-language, Kuratov et al. (2024), Yuan et al. (2024) and Levy et al. (2024) develop benchmarks to evaluate how well language models extract and reason over distributed facts in the presence of noise text. Hsieh et al. (2024) develop a rich set of tasks to evaluate long context performance with different dependency structures in natural language.

**Chain-of-thought prompting:** Wei et al. (2023) and Kojima et al. (2023) explore prompting strategies to induce multi-step reasoning in instruction tuned models. They find that models perform better on various tasks when prompted to output intermediate steps in the reasoning chain.

While most of the existing literature focuses on chat-style natural language generation, our work focuses on autocomplete-style code generation, where latency requirements often constrain the use of 'scratchpad' methods such as chain-of-thought to improve reasoning. We situate our work in between the PVR work of Zhang et al. (2022), which is highly controllable but uses a fairly abstract task, and the code-with-dependencies benchmarking work exemplified by Yu et al. (2024), which is more realistic but also makes it more difficult to run controlled experiments that allow discovery of specific failure modes. We design multi-step retrieval tasks that are similarly centered on indirection, but focus on evaluating inference-time limitations of pre-trained code models, and expand the input design space to include the effects of position and ordering, in addition to the number of hops.

---

[1]We open source the code to generate these tasks at https://github.com/apple/ml-key-retrieval-code-tasks.

# 3 Long Context Multi-Step Key Retrieval

## 3.1 Task Design

To study long-range dependency handling we propose four multi-step key retrieval tasks of increasing complexity (one-step, two-step, three-step, and concatenation retrieval). These extend the key-retrieval task in Rozière et al. (2024) and test models' ability to integrate multiple pieces of information spread throughout a long context window to make a completion.

<div>

(a) One Step

```
# ...
def key():
    return "xdfgew"
# ...
assert key() ==
```

(b) Two Step

```
# ...
def value():
    return "xdfgew"
# ...
def key():
    return value()
# ...
assert key() ==
```

(c) Three Step

```
# ...
def value_2():
    return "xdfgew"
# ...
def value_1():
    return value_2()
# ...
def key():
    return value_1()
# ...
assert key() ==
```

(d) Concatenation

```
# ...
def value_1():
    return "xdfgew"
# ...
def value_2():
    return "asdahj"
# ...
def key():
    return value_1() +
     value_2()
# ...
assert key() ==
```

</div>

Figure 1: Key retrieval tasks with increasing levels of difficulty. Function names and return values are randomized in the actual prompt. See Appendix G for a complete example.

**One-step retrieval** is equivalent to the key-retrieval task in Rozière et al. (2024). We differ from their design by using random strings to construct function names rather than fixed function names, and return values that are string literals rather than integer literals. Using a simple template system, we first generate a *key function* that returns a random string. The model is then asked to complete an *assert* statement on the return value of this function. All return strings that are 10 characters long and all function names are between 13 and 20 characters long.

In **two-step retrieval**, the key function calls a *value function* that returns the string. **Three-step retrieval** adds an additional function call between the key function and the value function that returns the string. In the **concatenation retrieval** task the key function calls two value functions and returns the concatenation of their returned strings Figure 1.

To turn each of these into a long context problem we insert varying amounts *irrelevant* code into the context window, we detail this in Section 3.3. In all cases the assert statement is on the result of the key function and is placed at the end of the prompt.

## 3.2 Avoiding Reliance on Parametric Knowledge and Trivial Solutions

To make sure the model is not solely relying on parametric knowledge (i.e., knowledge stored in the weights) to solve the task, we construct function names from two or three random sequences of lowercase characters or digits, joined with underscores (e.g., `zcxjdz_309521_xcdgfp`). Return values are random strings of lowercase characters (e.g., `pczjdfeyxc`). This makes it extremely unlikely that such functions exist in the model's training data, and therefore to complete the task successfully, the model must use the information from the prompt.

In our early experiments we observed that models may produce trivial solutions such as `assert foo() == foo()`. While these are technically correct , such solutions prevent us from testing the models' ability to retrieve information from the context. To prevent this, we *guide the decoding* of tokens to make sure that the response starts as a string literal[2]. As we sample output tokens, we use the `prefix_allowed_tokens` functionality in the Hugging Face transformers library (Wolf et al., 2020) to ensure that the output starts

---

[2]We are not able to do this guided decoding for the hosted GPT4o models, we instead add an opening quote to the prompt, i.e. `assert key_func() == "`.

with any number of spaces followed by a single or a double quote. Once an initial quote mark is produced, generation proceeds unrestricted.

### 3.3 Long Context Construction

To turn the tasks described above into long context inference problems we add irrelevant snippets to the context window from two sources. Motivated by findings in Min et al. (2019) and Chen & Durrett (2019) that show that the presence of good distractor functions are important when measuring multi-step reasoning, we first generate a number of synthetic *distractor* functions that use the same template as the the key and value functions that contain the task information. Then we sample standalone Python functions from the HumanEval dataset (Chen et al., 2021) to fill out the context window to our desired size. Algorithm 1 describes this dataset generation process in more detail.

---

**Algorithm 1** Generate Long Context Retrieval Tasks

---

$n_k \leftarrow$ number of unique key functions
$n_d \leftarrow$ number of synthetic distractor functions
$n_t \leftarrow$ maximum number of tokens
$n_p \leftarrow$ maximum number of position combinations
**repeat** $n_k$ **times**
    *snippets* $\leftarrow$ Generate key and value function(s) for task
    *assert* $\leftarrow$ Generate assert statement
    *irrelevant* $\leftarrow$ Generate $n_d$ distractor functions + randomly sampled functions from HumanEval such that $\text{TOKENCOUNT}(\textit{snippets} + \textit{irrelevant} + \textit{assert}) \lesssim n_t$
    *positions* $\leftarrow$ All $\text{LEN}(\textit{snippets})$ combinations of integers in $1 \ldots \text{LEN}(\textit{snippets} + \textit{irrelevant})$
                                          $\triangleright$ If $> n_p$ randomly sample $n_p$ of these combinations
    **for each** *position combination* in *positions* **do**
        **for each** permutation of *snippets* **do**
            *prompt* $\leftarrow$ *irrelevant*
            Insert each function in *snippets* into *prompt* using the positions in *position combination*
            Add *prompt* + *assert* to the list of generated prompts
**end**

---

This generates a dataset that allows us to compare the effect of the following variables: **position** of task related snippets within the input context, **relative order** of the task-related snippets, and the **spread** of task related snippets. We vary these factors while keeping the irrelevant functions used constant for each key function we generate. To aid reproducibility we fix the random seed during generation so that the prompt set can easily be re-generated.

In our experiments, we set $n_k$ to 100 for one-step task, and 20 for other tasks; $n_d$ to 0, 1, and 5; $n_p$ to $\infty$, 150, 50, 50 for one-step, two-step, three-step, and concatenation tasks respectively (this results in a maximum of 6000 prompts for each condition). We set $n_t$ to 2k, 4k and 8k tokens.

### 3.4 Models

We evaluate these tasks on six open source models: StarCoderBase-1B, StarCoderBase-7B, StarCoderBase-15.5B (Li et al., 2023), StarCoder2-7B (Lozhkov et al., 2024), Mistral-7B-v0.1 (Jiang et al., 2023) and DeepSeekCoder-6.7B-base(Guo et al., 2024). We also evaluate on GPT-4o-mini and GPT-4o (OpenAI, 2024). The StarCoder family of models are competitive on code generation benchmarks and available in a number of different parameter counts. StarCoder2-7B and DeepSeekCoder-6.7B-base models have been pre-trained on "repository level" data with the explicit aim of improving performance in contexts where there are dependencies between code units. Mistral is a general purpose LLM that is nonetheless competitive on code generation benchmarks. Similarly the GPT-4o models display strong performance on code generation

---

[2]Specifically gpt-4o-mini-2024-07-18 and gpt-4o-2024-11-20.

tasks even though they are not specific to code. We use implementations from the Hugging Face transformers library (Wolf et al., 2020) for the open source models.

Table 1: Models evaluated.

| Model | HumanEval[3] | Max Context Size[4] | Sliding Window Size |
|---|---|---|---|
| Mistral-7B | 30.5 | 8192 / 131k | 4096 |
| StarCoder2-7B | 35.4 | 16384 | 4096 |
| StarCoderBase-1B | 15.1 | 8192 | N/A |
| StarCoderBase-7B | 30.5 | 8192 | N/A |
| StarCoderBase-15.5B | 29.3 | 8192 | N/A |
| DeepSeekCoder-6.7B-base | 49.4 | 16384 | N/A |
| GPT-4o-mini | 87.2 | 128k | N/A |
| GPT-4o | 90.2 | 128k | N/A |

We are primarily interested in seeing how the performance of a given model *changes* as **task difficulty** (i.e., number of hops needed) and **context size** varies. We evaluate multiple models to see how consistent effects are across multiple models.

### 3.5 Results

#### 3.5.1 Overall Task Performance

Our primary metric is **accuracy@k**, which is similar to the **pass@k** metric introduced in Chen et al. (2021), the only difference is instead of running unit tests, we simply check if the string literal produced is the expected string. We compute **accuracy@3** over **10** generations for each input prompt. Hyperparameters for generation are in Appendix D

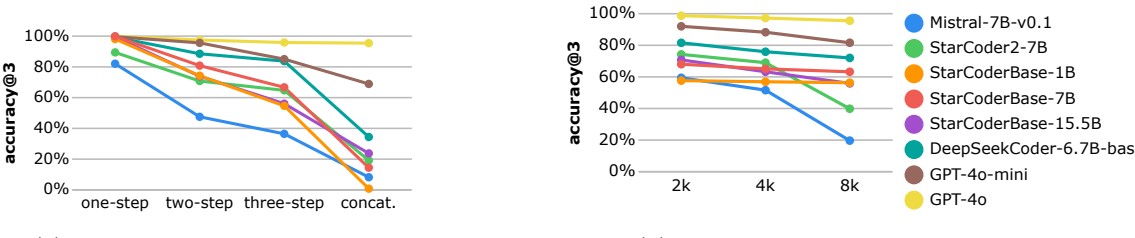

(a) Task performance by model.

(b) Task performance by prompt length.

Figure 2: Overall task performance. For detailed scores, see Table 5 in appendix.

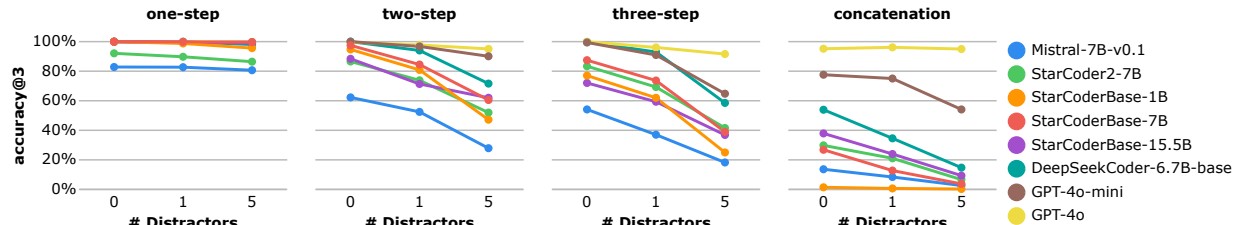

Figure 3: Effect of number of distractors by task variant. For detailed scores, see Table 6 in appendix.

**Task difficulty:** Figure 2a shows that tasks increase in difficulty in the following order: one-step, two-step, three-step, and concatenation. We note particularly weak performance on the concatenation task. Another

---

[2]HumanEval scores for StarCoder* models are from Lozhkov et al. (2024), StarcoderBase-1B scores are from `https://huggingface.co/bigcode/starcoderbase-1b`. Mistral-7B scores are from Jiang et al. (2023). GPT scores are from `https://openai.com/index/gpt-4o-mini-advancing-cost-efficient-intelligence/`

[3]Mistral-7B was trained with 8192 context size but reports support for up to 131k tokens.

factor affecting difficulty is the number of **distractor functions**. For the more difficult tasks we see a consistent degradation in performance as distractors are added (Figure 3).

**Context size:** In Figure 2b we observe a small performance drop as context sizes increases for StarCoderBase-1B, StarCoderBase-7B, StarCoderBase-15B, DeepSeekCoder-6.7B, GPT-4o-mini and GPT-4o. For StarCoder2-7B and Mistral-8B, which both use sliding window attention, a large drop in performance occurs when moving from 4k to 8k context length, we hypothesize that this is related to the size of the sliding window used and will explore this in more detail in Section 3.5.3.

### 3.5.2 Incorrect Responses

We analyzed the incorrect responses from models and note a number of distinct failure modes. Firstly a noticeable amount of incorrect generations are return values of distractor functions. Considering the 1 and 5 distractor conditions, approximately 10% of all responses are distractor answers across models (min=1.8%, max=16.2%). Within just the incorrect responses, approximately 20% are distractors (min=10.9%, max=33.4%). Full details are in Table 7 and Table 8 in the appendix.

For the concatenation task a common failure mode is returning a partial answer (i.e. one of the two values to be concatenated). Approximately 33.1% of all incorrect responses for this task are partial answers from one of the two strings value functions. See Table 9 for details.

To get further insight into other incorrect responses we also conduct an edit distance analysis. This allows us to see if models are mostly getting the answer right and maybe failing on just a few tokens. We compute Levenshtein distance between model responses and ground truth answers for the incorrect responses and find the mean distance to be 10.97. We note that the length of the expected string is 11 in all tasks. This indicates that when the model is wrong it is generally completely wrong rather than just incorrectly generating one or two tokens. Details of this analysis can be found in Table 10

### 3.5.3 Effect of Task Snippet Position

We focus the rest of our analysis on the five-distractor condition as it allows us to see a good variation in task performance.

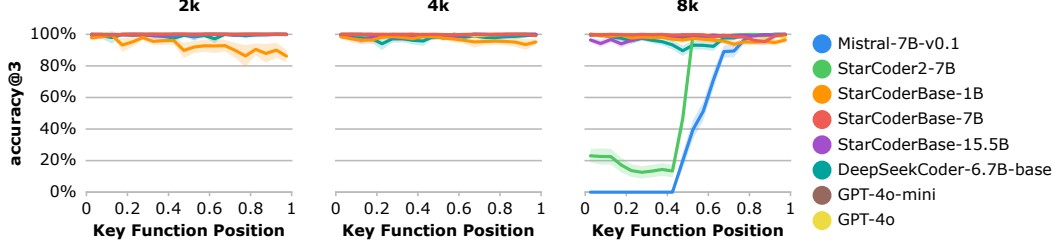

Figure 4: Effect of key function position on one-step task with 5 distractors. Position is defined as index of the first token in the function normalized by the total number of tokens in the prompt, and grouped into 20 bins. 0 is the beginning of the prompt and 1 is at the end. Shaded regions represent 95% confidence intervals

In the **one-step** retrieval task we see good performance with respect to *key function position* across the entire context window with two notable exceptions. At the 8k context size **Mistral-7B-v0.1** and **StarCoder2-7B** are unable to perform the one-step task when the key function is more than ~4k tokens away from the generation site Figure 4. Both models use a sliding window attention mechanism with a 4k window size. While this mechanism theoretically allows the model to scale very large context sizes (Jiang et al. (2023) reports a theoretical attention span of 131k tokens), our results indicate that the model fails to retrieve precise information at a distance greater than the sliding window.

**Mistral-7B-v0.1** allows for changing the sliding window size at run time. To futher confirm the result above, we measured one-step retrieval performance with sliding windows of 2048 and 8192 (in addition to the default 4096) at the 8k context size, and find that the drop in performance relative to key function

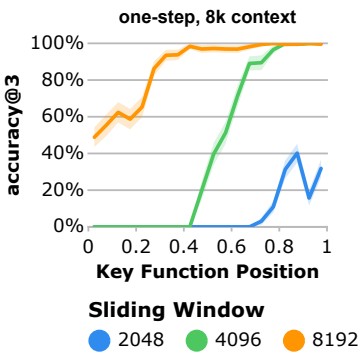

Figure 5: Varied sliding window sizes for Mistral-7B-v0.1 on the one-step task with 5 distractors.

position is generally consistent with this result. However we note that even at the 8192 sliding window size, performance begins to drops when the key function is approximately 60% of the size of the prompt away from the generation site, it however does not drop to zero as in the 2048 and 4096 case. Figure 5 shows this in detail.

The **two-step** task has two relevant task snippet positions, Figure 6 shows heatmaps of task performance vs. both key function position and value function position. We see that performance is worse when the key function *appears before* the value function in the prompt for the first two models. We call this a **forward reference** and it turns out this has a large effect on performance in most models. We also observe that performance is generally higher along the diagonal, i.e., when the functions are closer to each other. We elaborate on this further in Section 3.5.4

For **three-step** and **concatenation** tasks, it becomes harder to directly analyze position effects, as there are three task-relevant code snippets with independent positions. We instead explore performance by number forward references, and by spread between task-relevant snippets in the following sections.

### 3.5.4   Effect of Forward References

We define a **forward reference** as a function calling a not-yet-defined function (i.e. one that will be defined later in the prompt). We find that forward references have a negative impact on task performance in all models, in most conditions we tested. The exceptions to this are highlighted in blue in Table 2, and mainly occur in the three-step task where we sometimes observe an increase in performance when going from zero to one forward reference followed by a drop going from two to three forward references. Table 2 shows these results in detail. We see drops as large as 44.9% in models like StarCoderBase-7B (three-step task at 4k), 19.4% for GPT-4o-mini (concatenation task at 4k) and 12.3% for GPT4o (two-step task at 8k).

### 3.5.5   Effect of Task Snippet Spread

We saw in Figure 6 that the order of functions in multi-step tasks and the distance between functions affects performance. We define *spread* as the distance between the first token of the first task-related snippet and the last token of the last task-related snippet, *not including the assert statement* at the end of the prompt.

In the two-step task we broadly see three behaviors with respect to spread, the GPT4o models show little variation as spread increases. DeepSeekCoder-6.7B-base, StarCoder2-7B and Mistral-7B-v0.1 show performance drops at moderate amount of spreads but perform well when snippets are very close to each other or very far from each other. Snippets with maximal spread are located close to the beginning and end of the prompt and result in higher performance. The other models (StarCoderBase-1B, StarCoderBase-7B, and StarCoderBase-15B) show a drop in performance as spread increases, but only in the presence of forward references. Figure 7 shows the effect of task snippet spread for StarCoderBase-7B DeepSeekCoder-6.7B-base with 4k context length on the two-step retrieval task with results for the other models in the appendix (Figure 10).

**StarCoderBase-7B**

**StarCoder2-7B**

**GPT-4o-mini**

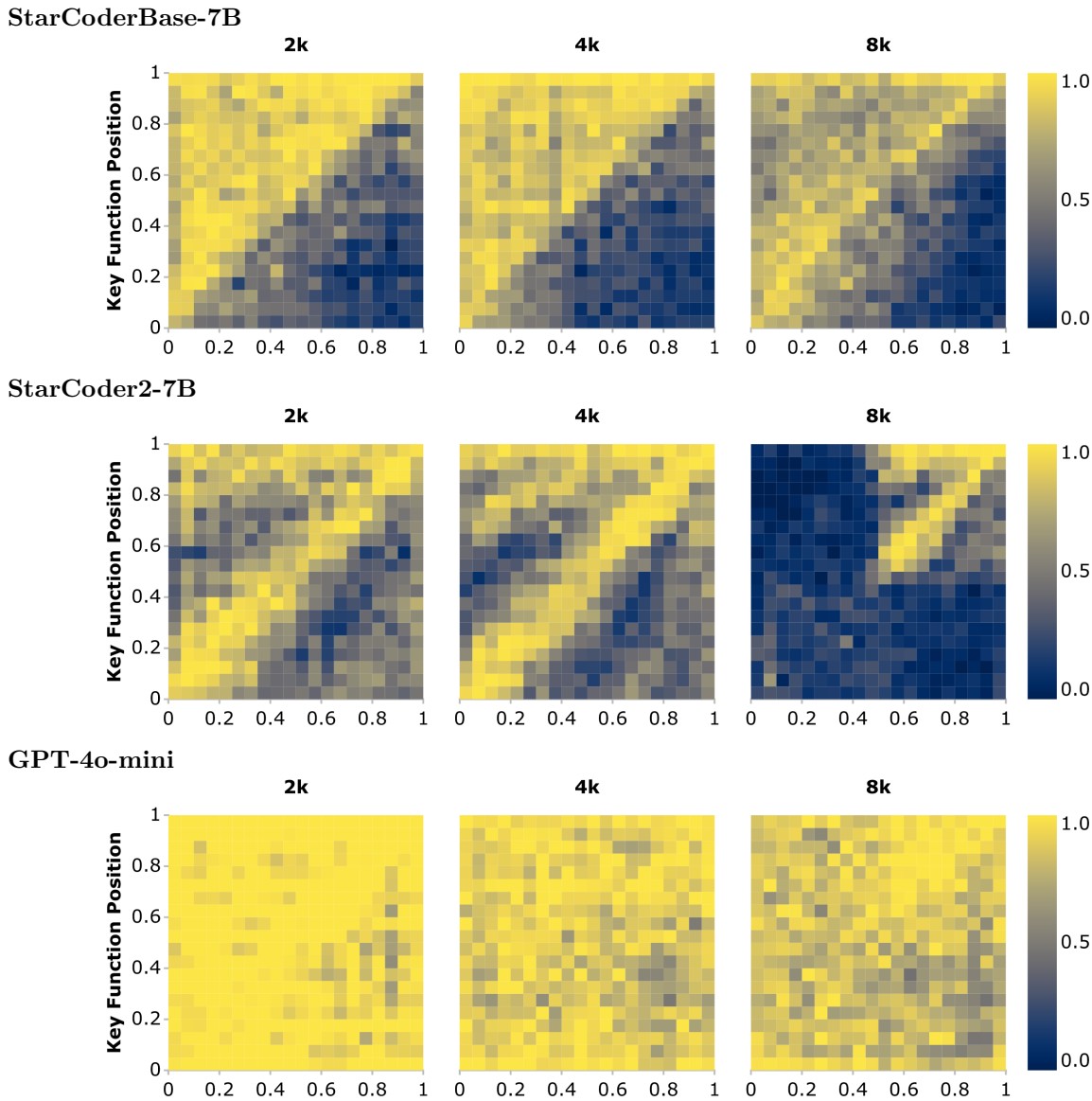

Figure 6: Effect of key and value function position for two-step task with 5 distractors. Top-to-bottom StarCoderBase-7B, StarCoder2-7B, GPT-4o-mini. Color represents accuracy@3 score.

For the three-step task similar, effects are clearly visible at 2k and 4k context size but are not as consistent across different models Figure 11. For the concatenation task, the effect is less pronounced as model's performance is generally low regardless of spread and the number of forward references and there is not much room for separation Figure 12.

# 4 Improving Retrieval Performance with Call Graph Comments

We saw in Section 3.5.4 that performance degrades for the multi-step tasks in the presence of forward references. Loosely inspired by chain-of-thought prompting (Wei et al., 2023; Kojima et al., 2023) that encourages models to produce intermediate token representing sub-parts of the problem, we considered if injecting information about function call relationships into the prompt would help performance. This allows us to control the amount of additional information about the problem provided and importantly, *where it*

Table 2: Model accuracy (%) vs. number of forward references. 5-distractor condition. 95% confidence interval shown adjacent. Performance generally drops as number of forward references increases. Blue indicates exceptions to this behavior, indicating a condition that performs worse than the condition with one more forward reference.

**Two-step Task**

| | 2k | | 4k | | 8k | |
|---|---|---|---|---|---|---|
| Context Size | | | | | | |
| # Forward References | 0 | 1 | 0 | 1 | 0 | 1 |
| Mistral-7B-v0.1 | $60.1_{\pm1.1}$ | $21.1_{\pm1.2}$ | $46.4_{\pm1.3}$ | $22.5_{\pm1.3}$ | $8.0_{\pm1.3}$ | $9.2_{\pm1.3}$ |
| StarCoder2-7B | $73.5_{\pm0.1}$ | $57.6_{\pm0.4}$ | $69.4_{\pm0.4}$ | $56.5_{\pm0.8}$ | $28.1_{\pm0.6}$ | $26.8_{\pm1.2}$ |
| StarCoderBase-1B | $55.3_{\pm0.4}$ | $39.4_{\pm0.7}$ | $49.7_{\pm1.0}$ | $33.9_{\pm1.1}$ | $65.6_{\pm1.1}$ | $39.3_{\pm1.3}$ |
| StarCoderBase-7B | $90.0_{\pm1.4}$ | $38.1_{\pm1.1}$ | $87.2_{\pm1.4}$ | $34.1_{\pm1.1}$ | $76.0_{\pm0.8}$ | $37.7_{\pm0.8}$ |
| StarCoderBase-15.5B | $85.5_{\pm1.3}$ | $63.4_{\pm1.4}$ | $82.3_{\pm1.4}$ | $49.2_{\pm1.4}$ | $63.4_{\pm1.4}$ | $28.3_{\pm1.3}$ |
| DeepSeekCoder-6.7B-base | $83.7_{\pm1.0}$ | $75.2_{\pm1.3}$ | $74.4_{\pm1.0}$ | $67.7_{\pm1.4}$ | $70.2_{\pm1.3}$ | $58.5_{\pm1.3}$ |
| GPT-4o-mini | $98.4_{\pm1.4}$ | $95.7_{\pm1.4}$ | $91.4_{\pm1.4}$ | $87.2_{\pm1.3}$ | $87.4_{\pm1.4}$ | $80.3_{\pm1.4}$ |
| GPT-4o | $99.8_{\pm0.8}$ | $98.3_{\pm1.3}$ | $98.6_{\pm0.9}$ | $93.6_{\pm1.3}$ | $96.2_{\pm1.2}$ | $83.9_{\pm1.3}$ |

**Three-step Task**

| | 2k | | | 4k | | | 8k | | |
|---|---|---|---|---|---|---|---|---|---|
| Context Size | | | | | | | | | |
| # Forward References | 0 | 1 | 2 | 0 | 1 | 2 | 0 | 1 | 2 |
| Mistral-7B-v0.1 | $28.1_{\pm2.1}$ | $24.4_{\pm1.0}$ | $16.8_{\pm2.2}$ | $19.8_{\pm2.3}$ | $22.3_{\pm1.1}$ | $16.9_{\pm2.1}$ | $2.7_{\pm2.3}$ | $11.7_{\pm1.2}$ | $10.3_{\pm2.1}$ |
| StarCoder2-7B | $61.5_{\pm1.1}$ | $60.5_{\pm0.5}$ | $45.7_{\pm1.8}$ | $46.2_{\pm1.9}$ | $48.1_{\pm0.8}$ | $37.5_{\pm2.0}$ | $14.4_{\pm2.0}$ | $23.1_{\pm0.8}$ | $16.0_{\pm1.9}$ |
| StarCoderBase-1B | $32.3_{\pm2.8}$ | $29.5_{\pm1.1}$ | $18.1_{\pm2.7}$ | $29.5_{\pm3.0}$ | $28.5_{\pm1.4}$ | $14.9_{\pm2.8}$ | $19.8_{\pm2.8}$ | $23.8_{\pm1.4}$ | $8.2_{\pm2.8}$ |
| StarCoderBase-7B | $63.4_{\pm2.0}$ | $41.3_{\pm0.9}$ | $18.8_{\pm1.7}$ | $65.4_{\pm1.8}$ | $42.0_{\pm0.9}$ | $20.5_{\pm1.6}$ | $41.9_{\pm0.8}$ | $35.6_{\pm0.8}$ | $15.9_{\pm1.3}$ |
| StarCoderBase-15.5B | $57.2_{\pm2.4}$ | $50.6_{\pm1.2}$ | $37.2_{\pm2.4}$ | $54.0_{\pm2.4}$ | $39.7_{\pm1.3}$ | $24.9_{\pm2.4}$ | $34.5_{\pm1.8}$ | $21.0_{\pm1.1}$ | $10.5_{\pm1.7}$ |
| DeepSeekCoder-6.7B-base | $69.4_{\pm2.4}$ | $73.1_{\pm1.2}$ | $56.2_{\pm2.3}$ | $52.2_{\pm2.3}$ | $61.3_{\pm1.2}$ | $52.6_{\pm1.9}$ | $42.9_{\pm2.2}$ | $50.5_{\pm1.0}$ | $40.1_{\pm1.3}$ |
| GPT-4o-mini | $68.0_{\pm2.2}$ | $81.8_{\pm1.1}$ | $70.9_{\pm1.9}$ | $53.3_{\pm2.3}$ | $67.1_{\pm1.1}$ | $62.8_{\pm1.8}$ | $39.4_{\pm2.0}$ | $56.3_{\pm1.1}$ | $50.4_{\pm1.3}$ |
| GPT-4o | $95.7_{\pm2.2}$ | $96.7_{\pm1.2}$ | $89.6_{\pm1.6}$ | $88.7_{\pm2.0}$ | $91.3_{\pm1.1}$ | $86.6_{\pm1.8}$ | $85.2_{\pm2.3}$ | $91.0_{\pm1.1}$ | $86.8_{\pm1.6}$ |

**Concatenation Task**

| | 2k | | | 4k | | | 8k | | |
|---|---|---|---|---|---|---|---|---|---|
| Context Size | | | | | | | | | |
| # Forward References | 0 | 1 | 2 | 0 | 1 | 2 | 0 | 1 | 2 |
| Mistral-7B-v0.1 | $11.8_{\pm1.8}$ | $4.3_{\pm1.5}$ | $3.1_{\pm1.3}$ | $2.4_{\pm1.2}$ | $0.6_{\pm0.9}$ | $0.6_{\pm0.8}$ | $0.8_{\pm1.0}$ | $0.2_{\pm0.6}$ | $0.2_{\pm0.5}$ |
| StarCoder2-7B | $17.7_{\pm0.4}$ | $14.7_{\pm0.6}$ | $7.9_{\pm0.6}$ | $6.4_{\pm0.6}$ | $5.4_{\pm0.7}$ | $4.2_{\pm0.8}$ | $1.8_{\pm0.8}$ | $0.7_{\pm0.9}$ | $0.7_{\pm1.2}$ |
| StarCoderBase-1B | $1.6_{\pm1.6}$ | $0.3_{\pm1.8}$ | $0.0_{\pm1.9}$ | $0.5_{\pm1.9}$ | $0.1_{\pm2.0}$ | $0.0_{\pm2.0}$ | $0.0_{\pm2.0}$ | $0.0_{\pm1.9}$ | $0.0_{\pm1.8}$ |
| StarCoderBase-7B | $13.9_{\pm1.1}$ | $9.3_{\pm0.7}$ | $4.0_{\pm0.5}$ | $2.0_{\pm0.5}$ | $1.6_{\pm0.2}$ | $1.2_{\pm0.2}$ | $0.6_{\pm0.3}$ | $0.6_{\pm0.1}$ | $0.3_{\pm0.2}$ |
| StarCoderBase-15.5B | $22.9_{\pm1.3}$ | $14.2_{\pm1.3}$ | $11.0_{\pm0.9}$ | $10.1_{\pm0.9}$ | $6.1_{\pm0.8}$ | $5.8_{\pm0.7}$ | $6.8_{\pm0.5}$ | $4.5_{\pm0.3}$ | $2.6_{\pm0.3}$ |
| DeepSeekCoder-6.7B-base | $40.3_{\pm1.5}$ | $24.7_{\pm1.2}$ | $20.2_{\pm1.1}$ | $14.2_{\pm1.1}$ | $8.5_{\pm0.8}$ | $7.2_{\pm0.8}$ | $9.6_{\pm0.8}$ | $4.6_{\pm0.7}$ | $3.2_{\pm0.5}$ |
| GPT-4o-mini | $79.8_{\pm0.4}$ | $71.2_{\pm0.2}$ | $63.6_{\pm0.0}$ | $64.9_{\pm0.2}$ | $52.7_{\pm0.1}$ | $45.5_{\pm0.0}$ | $46.1_{\pm0.0}$ | $35.4_{\pm0.0}$ | $27.7_{\pm0.0}$ |
| GPT-4o | $98.5_{\pm1.2}$ | $96.7_{\pm1.0}$ | $97.3_{\pm0.6}$ | $97.4_{\pm0.5}$ | $96.0_{\pm0.4}$ | $94.9_{\pm0.3}$ | $93.6_{\pm0.2}$ | $91.7_{\pm0.2}$ | $88.9_{\pm0.1}$ |

*appears.* So unlike chain-of-thought prompting, we do not allow the model to produce extra tokens but rather inject 'extra' tokens into particular areas of the prompt.

We focused on our most difficult tasks, namely the three-step and concatenation retrieval task with 5 distractors using the same models as in our previous experiment.

## 4.1 Experiment Setup

We add comments above each function that contain a lightweight description of the *call graph* associated with that function. We can construct these comments by annotating which functions are **"called by"** other functions, or annotating which functions **"call"** a given function, or a combination of **both**. In the three-step retrieval task we can also do this transitively and add annotation of *all the functions* that will be called to compute the result of a given function (as opposed to just the next function in the chain).

We considered two templates for these comments, in one variation ("names only") we simply included a comma-separated list of function names *and no other information*. This has the effect of introducing the tokens for the referenced function before it appears as a call target, though without any indication that it is even a function name.

**StarCoderBase-7B**

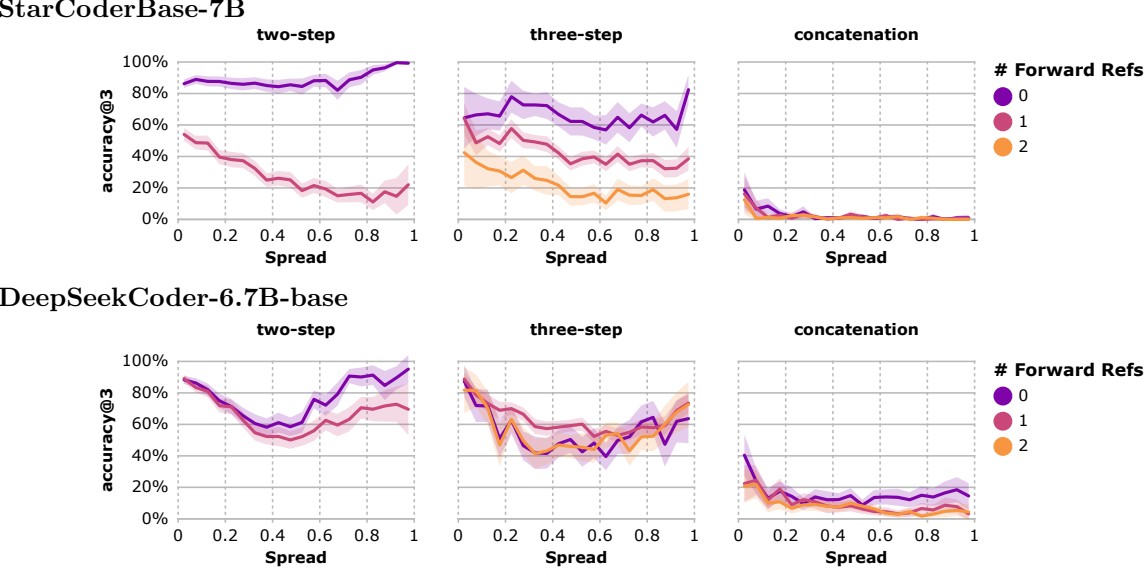

**DeepSeekCoder-6.7B-base**

Figure 7: Effect of spread (normalized to the number of tokens in the context) for StarCoderBase-7B (top) and DeepSeekCoder-6.7B-base (bottom) with 4k context size, 5-distractor condition. Shaded regions represent 95% confidence interval. For data on other models and context sizes, see Figure 10, Figure 11, and Figure 12.

In the other template ("full sentence") we use the phrases: *"This function is called by "* and *"This function calls "* followed by the comma-separated list of function names. See Appendix H for examples.

### 4.2 Results

Table 3: Call graph comment performance across all models. Accuracy@3

|  | | Comment Type | | | |
|---|---|---|---|---|---|
|  | Template | none | calls | called-by | both |
| Three-step | Names Only | 46.9 | 47.8 | 72.9 | 69.3 |
|  | Full Sentence | 46.9 | 55.3 | 74.2 | **76.8** |
| Concatenation | Names Only | 23.3 | 24.6 | 29.5 | 29.5 |
|  | Full Sentence | 23.3 | 27.6 | 32.8 | **36.4** |

We observe that the addition of call-graph comments has a positive effect on task performance. Table 3 shows a ~1.5x improvement on the concatenation task, and a ~1.6x improvement on the three-step retrieval task with the addition of call-graph comments aggregated across models. We do note that StarCoderBase-1B sees no improvement on the concatenation task. See Appendix C for detailed results.

**Full sentence vs. function names only:** The *full-sentence* template performs better than the *names-only* version (Table 3). This suggests that the models are able to take some advantage of the more complete natural language description and that there may be room for further improvement by tuning the template. However it is notable that simply including the names of the referenced functions before their function definition has such a large performance boost. We focus the rest of our analysis on the full-sentence condition as it was the best performing template.

Table 4: Call graph comment performance by depth across all models. Accuracy@3

| # Forward References | Comment Type | | |
|---|---|---|---|
|  | None | Next Hop | Full |
| 0 | 48.6 | 70.4 | **73.9** |
| 1 | 48.8 | 71.9 | **79.0** |
| 2 | 37.8 | 58.9 | **71.0** |

**Call graph directions:** Table 3 shows that while most of the benefit comes from the "X is *called by* Y" type of comment, using both directions produces the best improvement overall.

**Full graph vs. next-hop comments:** We noted earlier that in the three-step case we add comments describing the full call graph (i.e., up to two steps away). We experimented with only adding comments about the *next function in the call graph* and found that the full graph version performs best (Table 4).

**StarCoderBase-7B**

**GPT-4o-mini**

Figure 8: Effect of call graph comments on StarcoderBase-7B (top) and GPT-4o-mini (bottom) performance across context sizes, spread of task snippets and number of forward references. Shaded regions represent 95% confidence interva

**Context size, spread, and forward references:** Figure 8 shows that call-graph comments improve performance across all context lengths, number of forward references, and task-snippet spreads for StarCoderBase-7B (this trend holds for other models where the call graph comments make a difference). Full plots for all the models are found in Appendix C. We note that with respect to *spread* for the concatenation task, most of the improvement appears when the task-relevant snippets are closer together.

## 5    Discussion

Our findings align with those of Yu et al. (2024), who show that over 70% of functions in popular open-source repositories are "non-standalone" and report a 48% average performance drop when models are evaluated on such functions. Their report underscores the need to evaluate the ability of models to reason over interdependent code units. Our use of synthetic code enables us to expand the context beyond the 1K token limit in their benchmark and gain fine-grain control over the relations being tested.

Our results also help distinguish between *long-context support* and the ability to handle *long-range dependencies*. We find that models were able to perform well at the simpler one-step recall task over their entire context lengths (Figure 2a), but performance degrades as the number of steps required increases. This is further exacerbated by the presence of distractors which we believe are important for appropriately pressure testing the ability of models to do the required reasoning (Figure 3).

We also observe architectural choices that appear to make a tradeoff between efficient support of long-context inputs and capturing long-range token relationships. This is illustrated by models like Mistral and StarCoder2, whose use of sliding window attention greatly expands their supported context sizes, however our experiments show that when distance between relevant snippets is greater than the sliding window size used by these models, performance degrades quickly. Our tasks require *precise* recall over a long distance and it appears to be difficult to pass on information from earlier sliding windows in a way that allows successful completion of the task. As different approaches to scaling attention to larger context windows proliferate, practitioners should take care to understand and mitigate these effects if their task relies on precise recall of long-range dependencies. In the context of code generation, builders of retrieval augmented generation systems may find it desirable to rewrite code to bring code snippets that depend on each other closer together.

Our findings shed light on specific factors influencing the fragility of multi-step reasoning in large language models, including the presence of distractors, function ordering and architectural constraints such as sliding-window attention.

We note that models we tested struggled even more in the presence of forward references (Table 2). In this situation *all the information in the prompt is the same, but is presented in a different order*. Humans and compilers are able to resolve this problem and it would be ideal if models could do the same. While we do not know what causes this behavior, one hypothesis is that the causal attention used during training favors attending backwards in the context window and propagating information from earlier to later tokens.

While we leave the discovery of the mechanisms that underlie such behavior to future work, we do observe useful progress in the field that could be brought to bear. While there is still debate on whether examining attention patterns in model can provide faithful explanations (Jain & Wallace, 2019; Wiegreffe & Pinter, 2019), recent progress in mechanistic interpretability is developing tools to closely examine components involved in resolving relationships between tokens and the mechanisms transformers use to encode them (e.g. Elhage et al. (2021)). Hernandez et al. (2024) find that a subset of natural-language entity relations can be modeled with a single linear transformation. Brinkmann et al. (2024) analyze mechanisms involved in symbolic multi-step reasoning in transformers using a synthetic tree-traversal task and describe mechanism such as "backward-chaining" and "one-step" lookahead in the models they train. Such techniques could be applied to larger pre-trained models to understand the failure modes in resolving forward references that we observed in this work.

Finally we find that we can add information to the prompt that may short-circuit whichever mechanisms are involved in resolving function references. We see that introducing just the tokens of referenced functions before they are used in a function definition greatly improves performance on the more difficult multi-step tasks in all cases, especially where forward references are present (Table 3). We believe these scenarios provide a rich testbed to better examine how information flows across tokens particularly in scenarios where models must 'plan' for the future.

## 6 Limitations

Our experiments uses synthetically generated functions that are not as realistic as real code in the wild. Our tests only use top level functions composed via function call chains and concatenation. Nor do we test more complex structures of indirection such as recursion, branching control flow, or abstractions that cause functions in the call graph to be reused (e.g. helper functions used by multiple functions). These may introduce additional retrieval challenges or sensitivity to ordering not captured by our current setup. Furthermore our use of randomly generated function names results in a setting that is likely out-of-domain. Models may perform better with more realistic identifiers drawn from natural codebases.

We also evaluate only decoder-only causal transformers. Models with recurrence or bidirectional attention may behave differently and merit separate study, particularly with regard to the forward reference finding.

Suggestions such as re-ordering functions to bring dependent code closer together or adding annotations like call-graph comments would need to be tested in real retrieval-augmented code generation environments and should be seen as exploratory; we are not recommending these approaches over others that may improve performance on these tasks (e.g. fine tuning).

## 7 Acknowledgements

We thank Alan Leung, Titus Barik, Dominik Moritz, Samira Abnar, Forrest Huang, Jeff Nichols and Barry-John Theobald for their helpful feedback and support.

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

# A  Appendix

## A  Detailed Results: Experiment 1

We include detailed tables of accuracy@3 scores (percent scale). Table 5 shows overall scores by task and context length (2k, 4k, and 8k); Table 6 shows scores by the number of distractors; and Table 2 shows scores by the number of forward references.

Table 5: Overall accuracy@3 scores for each model, task, and context size. Each score in the context length columns is the mean across all number of distractors. The "mean" column shows the average score for the three context sizes.

| Task | Model | 2k | 4k | 8k | mean |
|------|-------|-----|-----|-----|------|
| one-step | Mistral-7B-v0.1 | 99.8±0.6 | 99.8±0.6 | 46.4±0.5 | 82.0 |
| | StarCoder2-7B | 100.0±0.2 | 99.8±0.3 | 68.1±0.3 | 89.3 |
| | StarCoderBase-1B | 97.1±0.5 | 98.3±0.6 | 98.5±0.7 | 98.0 |
| | StarCoderBase-7B | 100.0±0.4 | 99.9±0.3 | 99.6±0.1 | 99.8 |
| | StarCoderBase-15.5B | 99.8±0.6 | 99.5±0.5 | 99.0±0.3 | 99.5 |
| | DeepSeekCoder-6.7B-base | 99.7±0.6 | 99.2±0.5 | 98.8±0.4 | 99.2 |
| | GPT-4o-mini | 99.9±0.1 | 99.8±0.1 | 99.3±0.0 | 99.6 |
| | GPT-4o | 100.0±0.5 | 99.9±0.4 | 99.6±0.3 | 99.9 |
| two-step | Mistral-7B-v0.1 | 63.1±0.1 | 58.3±0.2 | 16.7±0.2 | 46.1 |
| | StarCoder2-7B | 81.9±0.0 | 82.6±0.1 | 46.4±0.1 | 70.3 |
| | StarCoderBase-1B | 73.3±0.1 | 73.5±0.1 | 73.7±0.1 | 73.5 |
| | StarCoderBase-7B | 81.6±0.1 | 80.1±0.1 | 79.2±0.9 | 80.3 |
| | StarCoderBase-15.5B | 82.2±0.0 | 74.7±0.1 | 64.0±0.8 | 73.7 |
| | DeepSeekCoder-6.7B-base | 90.1±0.1 | 88.0±0.1 | 85.6±0.1 | 87.9 |
| | GPT-4o-mini | 98.8±0.3 | 95.8±0.2 | 92.2±0.1 | 95.6 |
| | GPT-4o | 99.1±0.0 | 97.9±0.0 | 95.5±0.1 | 97.5 |
| three-step | Mistral-7B-v0.1 | 53.1±0.4 | 41.5±0.4 | 14.7±0.5 | 36.5 |
| | StarCoder2-7B | 78.9±0.2 | 75.3±0.3 | 40.0±0.3 | 64.7 |
| | StarCoderBase-1B | 55.9±0.4 | 55.2±0.5 | 52.9±0.6 | 54.7 |
| | StarCoderBase-7B | 67.0±0.6 | 66.9±0.5 | 66.2±0.4 | 66.7 |
| | StarCoderBase-15.5B | 65.8±0.5 | 57.1±0.5 | 45.3±0.6 | 56.1 |
| | DeepSeekCoder-6.7B-base | 88.3±0.6 | 83.9±0.6 | 79.0±0.6 | 83.7 |
| | GPT-4o-mini | 90.8±0.6 | 86.5±0.6 | 77.8±0.6 | 85.0 |
| | GPT-4o | 97.9±0.6 | 95.6±0.5 | 94.1±0.6 | 95.9 |
| concatenation | Mistral-7B-v0.1 | 17.3±0.4 | 6.7±0.4 | 0.7±0.4 | 8.2 |
| | StarCoder2-7B | 34.7±0.1 | 18.3±0.2 | 4.5±0.3 | 19.2 |
| | StarCoderBase-1B | 1.7±0.2 | 0.6±0.3 | 0.1±0.4 | 0.8 |
| | StarCoderBase-7B | 21.9±0.7 | 13.4±0.6 | 8.0±0.5 | 14.4 |
| | StarCoderBase-15.5B | 34.4±0.5 | 21.6±0.5 | 15.2±0.6 | 23.7 |
| | DeepSeekCoder-6.7B-base | 46.2±0.5 | 32.5±0.5 | 24.6±0.6 | 34.4 |
| | GPT-4o-mini | 78.6±0.6 | 70.9±0.5 | 57.2±0.5 | 68.9 |
| | GPT-4o | 97.9±0.5 | 95.5±0.5 | 92.9±0.5 | 95.4 |

Table 6: Overall model performance vs. number of distractors.

**One-step Task**

| Context Size | | | 2k | | | 4k | | | 8k |
|---|---|---|---|---|---|---|---|---|---|
| # Distractors | 0 | 1 | 5 | 0 | 1 | 5 | 0 | 1 | 5 |
| Mistral-7B-v0.1 | 100.0 | 99.9 | 99.6 | 100.0 | 100.0 | 99.5 | 48.5 | 48.2 | 42.8 |
| StarCoder2-7B | 100.0 | 100.0 | 99.9 | 100.0 | 100.0 | 99.6 | 76.3 | 69.3 | 59.8 |
| StarCoderBase-1B | 100.0 | 99.1 | 93.2 | 99.9 | 98.4 | 96.7 | 99.9 | 98.8 | 97.0 |
| StarCoderBase-7B | 100.0 | 100.0 | 99.9 | 100.0 | 100.0 | 99.9 | 100.0 | 99.7 | 99.1 |
| StarCoderBase-15.5B | 99.9 | 99.8 | 99.8 | 100.0 | 99.5 | 99.0 | 99.8 | 99.3 | 98.1 |
| DeepSeekCoder-6.7B-base | 100.0 | 100.0 | 99.3 | 100.0 | 100.0 | 98.0 | 100.0 | 99.7 | 96.8 |
| GPT-4o-mini | 100.0 | 99.6 | 100.0 | 100.0 | 99.7 | 99.7 | 100.0 | 98.7 | 99.2 |
| GPT-4o | 100.0 | 100.0 | 100.0 | 100.0 | 100.0 | 99.9 | 99.9 | 99.7 | 99.4 |

**Two-step Task**

| Context Size | | | 2k | | | 4k | | | 8k |
|---|---|---|---|---|---|---|---|---|---|
| # Distractors | 0 | 1 | 5 | 0 | 1 | 5 | 0 | 1 | 5 |
| Mistral-7B-v0.1 | 87.4 | 74.5 | 40.6 | 76.7 | 63.9 | 34.4 | 22.7 | 18.9 | 8.6 |
| StarCoder2-7B | 99.2 | 85.5 | 65.5 | 97.9 | 86.8 | 63.0 | 62.6 | 49.1 | 27.5 |
| StarCoderBase-1B | 98.5 | 80.9 | 47.3 | 96.2 | 82.5 | 41.8 | 89.5 | 79.2 | 52.4 |
| StarCoderBase-7B | 99.2 | 86.3 | 64.0 | 97.9 | 81.9 | 60.6 | 95.4 | 85.3 | 56.8 |
| StarCoderBase-15.5B | 93.6 | 80.9 | 74.4 | 89.6 | 68.8 | 65.8 | 82.0 | 64.3 | 45.8 |
| DeepSeekCoder-6.7B-base | 100.0 | 96.5 | 79.4 | 100.0 | 92.9 | 71.0 | 100.0 | 92.6 | 64.4 |
| GPT-4o-mini | 100.0 | 99.4 | 97.1 | 100.0 | 98.1 | 89.3 | 100.0 | 92.7 | 83.9 |
| GPT-4o | 100.0 | 98.2 | 99.1 | 100.0 | 97.6 | 96.1 | 99.0 | 97.4 | 90.0 |

**Three-step Task**

| Context Size | | | 2k | | | 4k | | | 8k |
|---|---|---|---|---|---|---|---|---|---|
| # Distractors | 0 | 1 | 5 | 0 | 1 | 5 | 0 | 1 | 5 |
| Mistral-7B-v0.1 | 81.4 | 54.0 | 23.8 | 61.4 | 42.3 | 21.0 | 19.5 | 14.7 | 10.0 |
| StarCoder2-7B | 95.6 | 83.0 | 58.2 | 94.5 | 85.4 | 46.0 | 60.0 | 39.5 | 20.5 |
| StarCoderBase-1B | 80.7 | 59.0 | 28.1 | 70.9 | 68.3 | 26.4 | 79.6 | 58.7 | 20.5 |
| StarCoderBase-7B | 91.7 | 68.0 | 41.2 | 82.6 | 75.9 | 42.3 | 88.0 | 77.1 | 33.4 |
| StarCoderBase-15.5B | 81.4 | 66.5 | 49.5 | 69.7 | 62.1 | 39.6 | 65.0 | 49.5 | 21.5 |
| DeepSeekCoder-6.7B-base | 100.0 | 95.3 | 69.7 | 99.9 | 93.6 | 58.3 | 99.3 | 90.0 | 47.5 |
| GPT-4o-mini | 100.0 | 94.8 | 77.7 | 99.8 | 95.7 | 64.1 | 98.5 | 82.3 | 52.5 |
| GPT-4o | 100.0 | 98.4 | 95.3 | 100.0 | 96.7 | 90.1 | 99.8 | 93.1 | 89.4 |

**Concatenation Task**

| Context Size | | | 2k | | | 4k | | | 8k |
|---|---|---|---|---|---|---|---|---|---|
| # Distractors | 0 | 1 | 5 | 0 | 1 | 5 | 0 | 1 | 5 |
| Mistral-7B-v0.1 | 27.0 | 18.5 | 6.4 | 12.8 | 6.2 | 1.2 | 1.1 | 0.6 | 0.4 |
| StarCoder2-7B | 54.3 | 36.2 | 13.4 | 26.5 | 23.0 | 5.4 | 8.6 | 3.9 | 1.1 |
| StarCoderBase-1B | 2.9 | 1.5 | 0.6 | 1.1 | 0.5 | 0.2 | 0.3 | 0.1 | 0.0 |
| StarCoderBase-7B | 37.5 | 19.2 | 9.1 | 26.9 | 11.7 | 1.6 | 16.1 | 7.3 | 0.5 |
| StarCoderBase-15.5B | 50.5 | 36.6 | 16.1 | 35.6 | 21.9 | 7.3 | 27.5 | 13.3 | 4.6 |
| DeepSeekCoder-6.7B-base | 60.1 | 50.1 | 28.4 | 54.2 | 33.3 | 10.0 | 47.4 | 20.4 | 5.8 |
| GPT-4o-mini | 85.4 | 78.8 | 71.5 | 74.4 | 84.0 | 54.4 | 73.0 | 62.2 | 36.4 |
| GPT-4o | 97.6 | 98.7 | 97.5 | 93.2 | 97.3 | 96.1 | 94.8 | 92.4 | 91.4 |

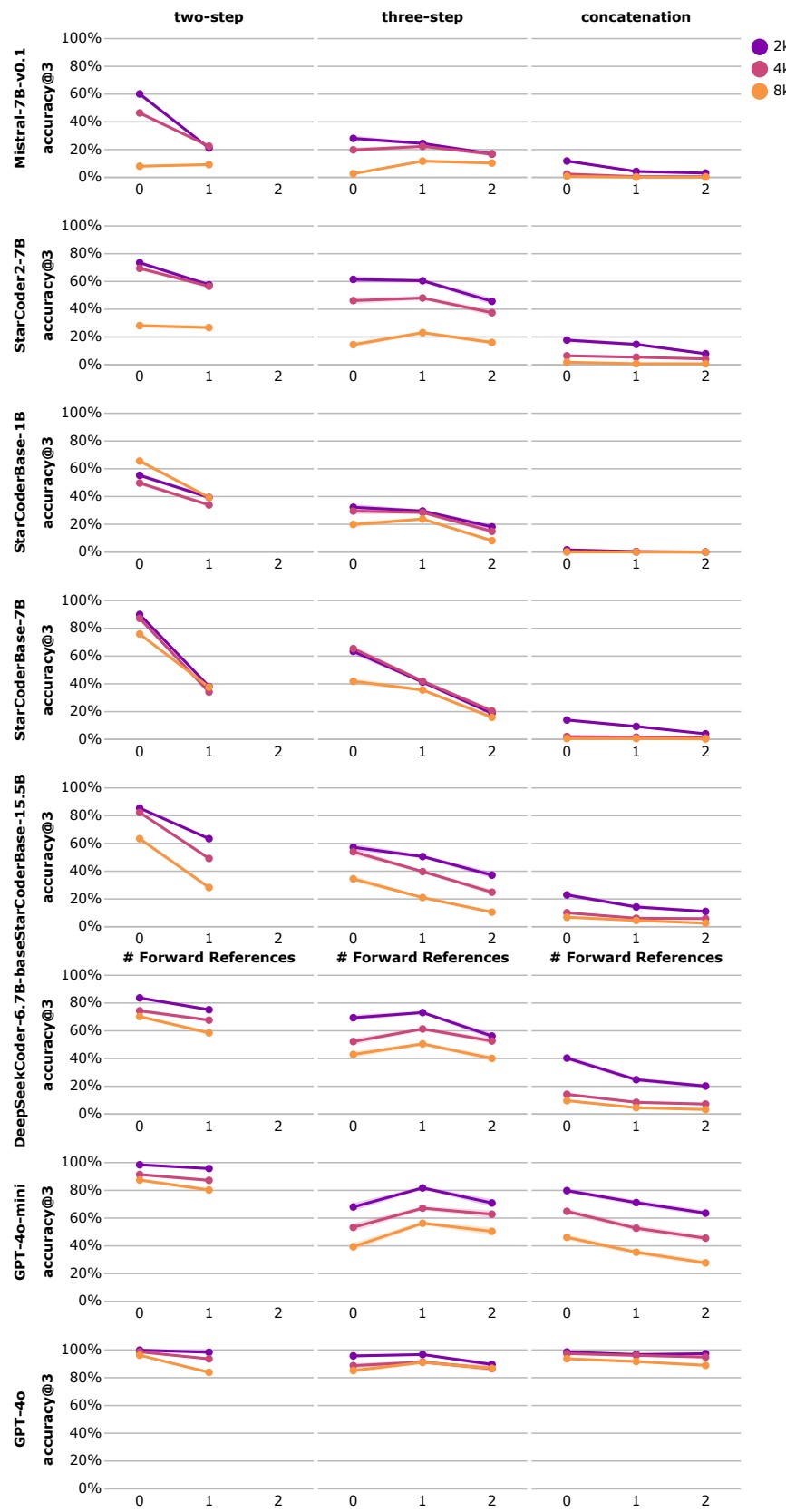

Figure 9: Effect of forward references by task variant, with 5 distractors. The data is showin Table 2

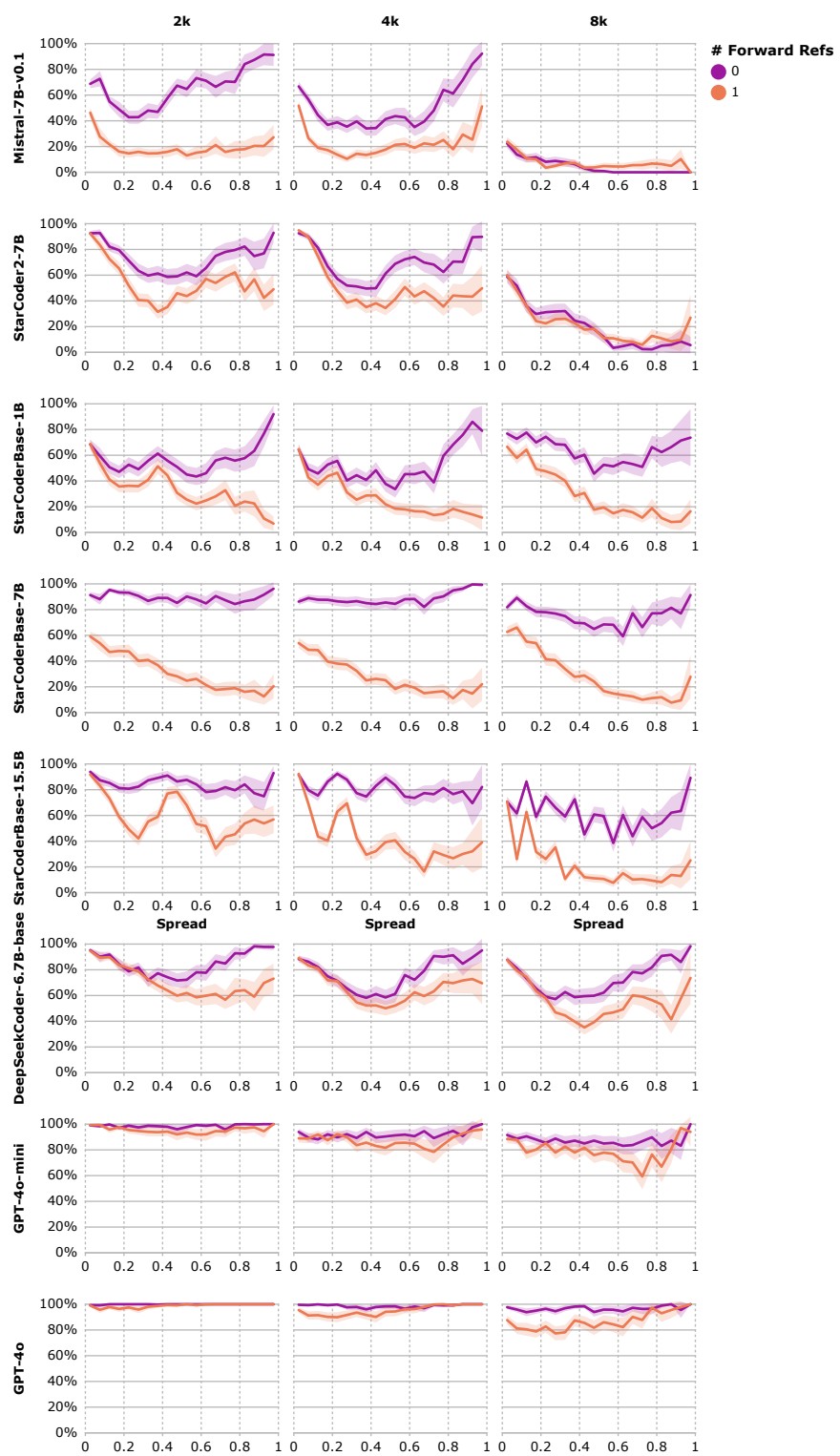

Figure 10: Effect of spread for two step tasks for each model, with 5 distractors.

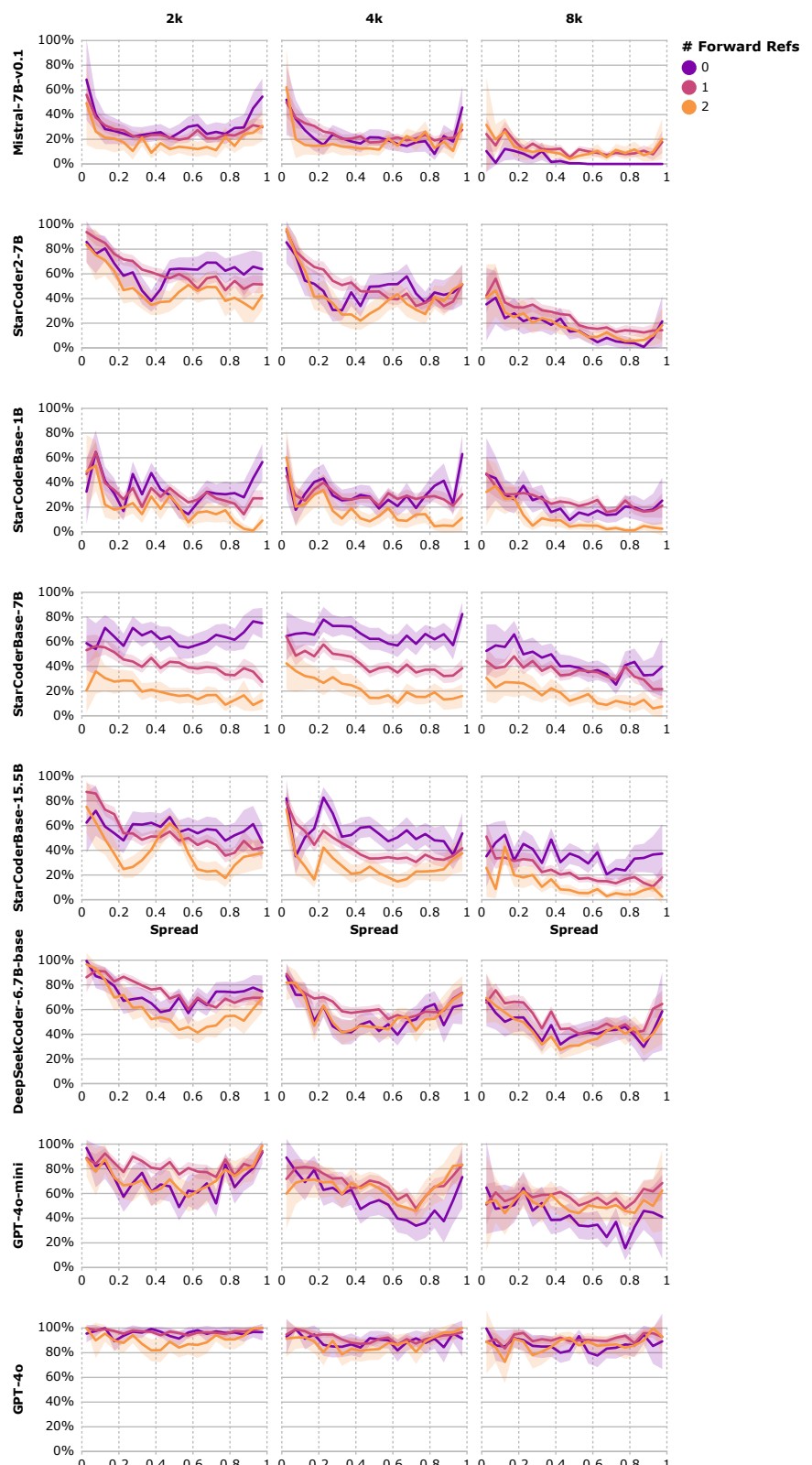

Figure 11: Effect of spread for three step tasks for each model, with 5 distractors.

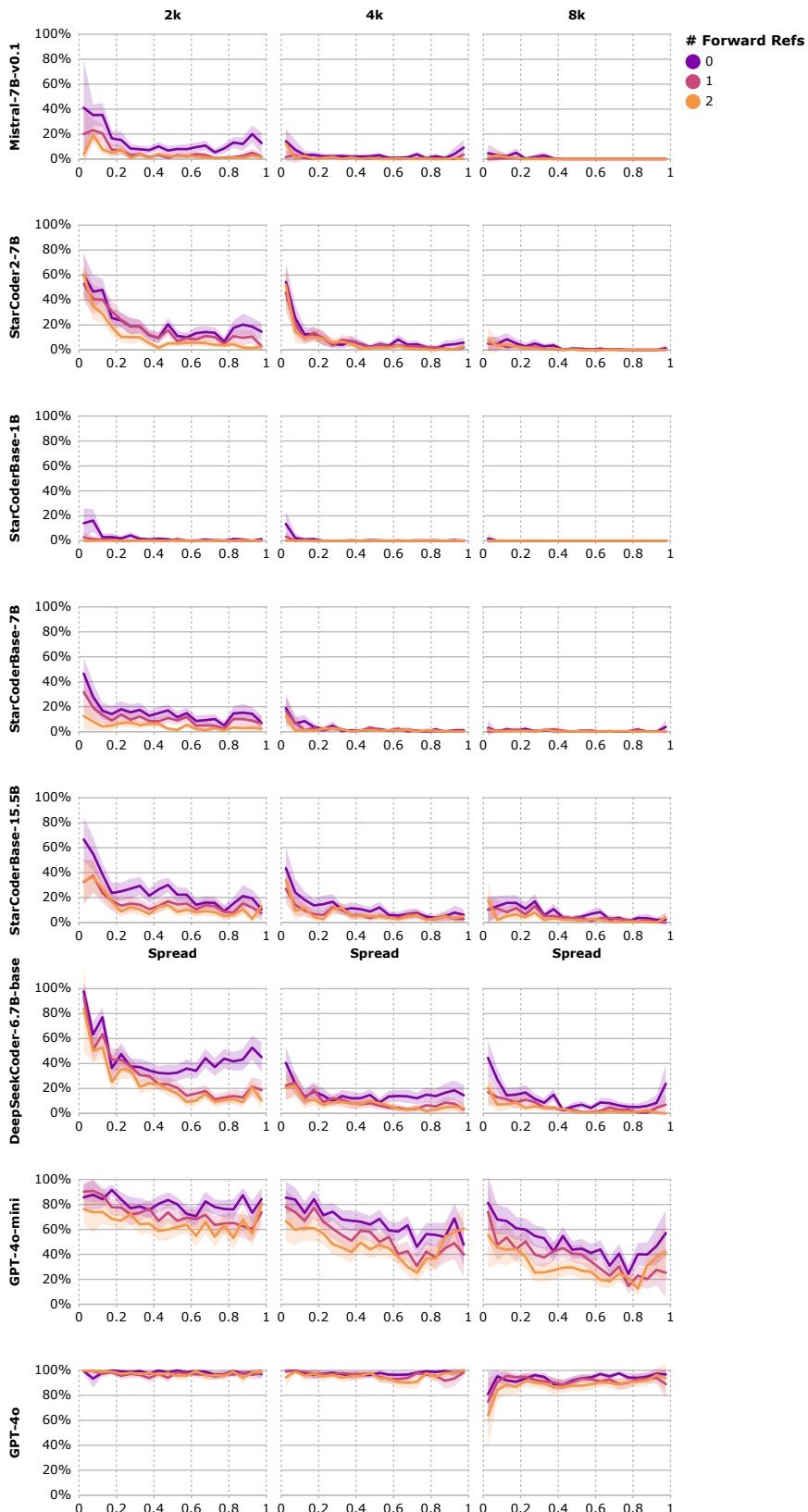

Figure 12: Effect of spread for concatenation tasks for each model, with 5 distractors.

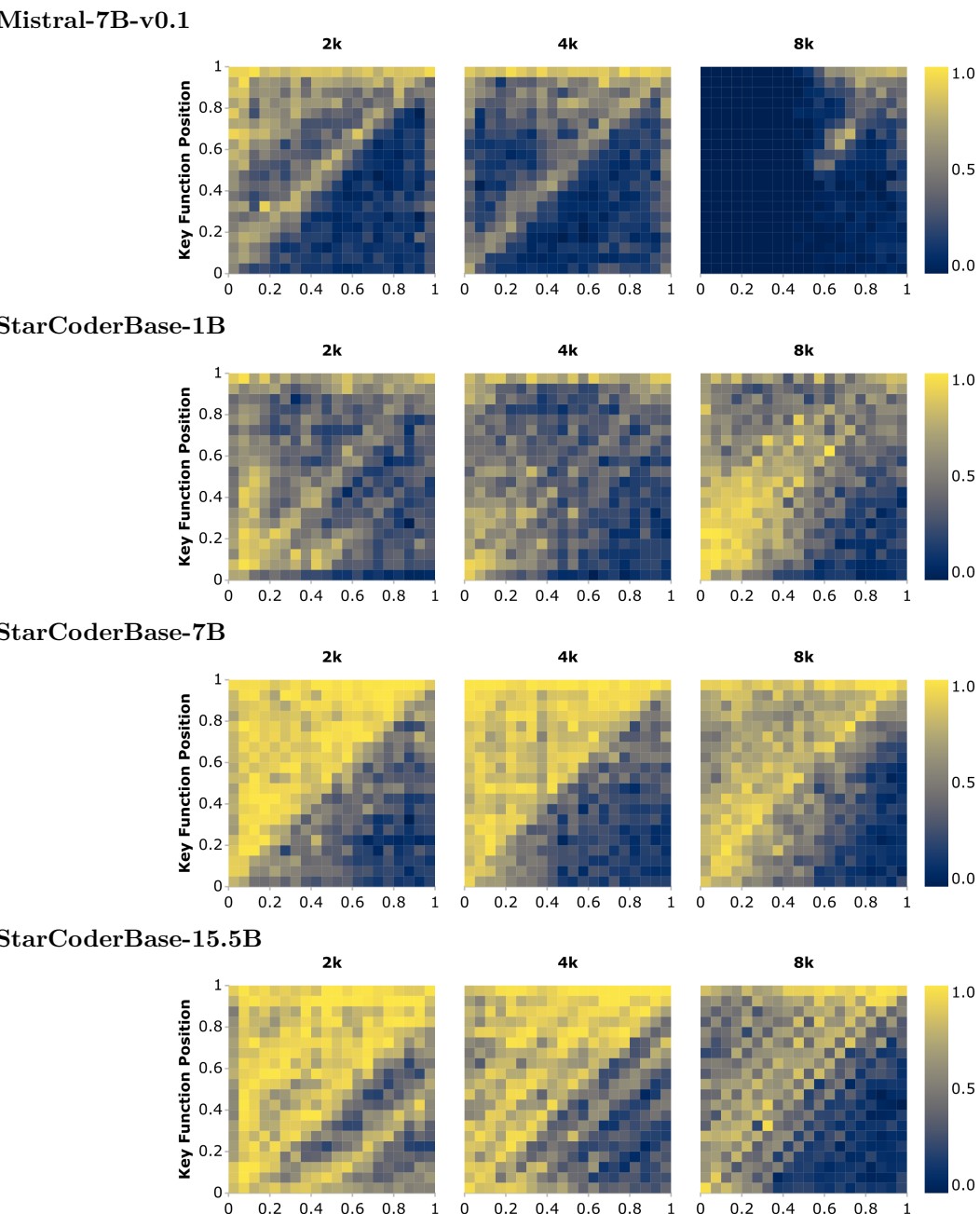

Figure 13: Effect of key and value function position for two-step tasks, with 5 distractors. Color represents accuracy@3 score.

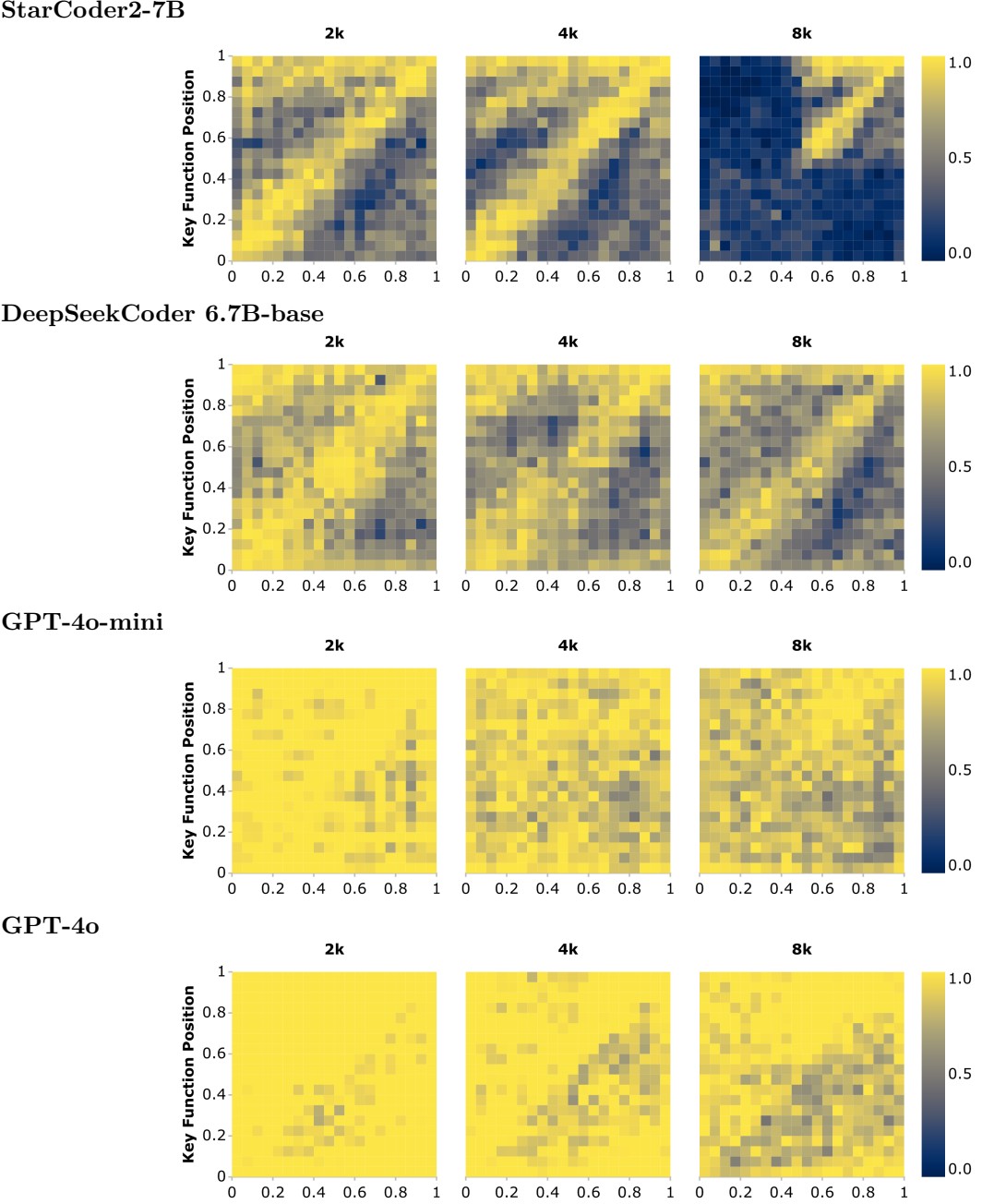

Figure 14: Effect of key and value function position for two-step tasks, with 5 distractors. Color represents accuracy@3 score.

# B    Incorrect Response Analysis

Here we present analyses of incorrect responses broken down by model. We also present details of the edit distance analysis.

Table 7: Percentage of distractor answers across all experiments with distractors

| Prompt Size | 2k | 4k | 8k | Mean |
|---|---|---|---|---|
| Mistral-7B-v0.1 | 10.5 | 9.4 | 5.3 | 8.4 |
| StarCoder2-7B | 16.7 | 19.4 | 12.9 | 16.3 |
| StarCoderBase-1B | 11.5 | 11.0 | 14.2 | 12.2 |
| StarCoderBase-7B | 7.7 | 9.5 | 11.7 | 9.6 |
| StarCoderBase-15.5B | 11.5 | 11.3 | 10.1 | 11.0 |
| DeepSeekCoder-6.7B-base | 16.0 | 17.5 | 15.1 | 16.2 |
| GPT-4o-mini | 5.2 | 7.3 | 11.1 | 7.9 |
| GPT-4o | 0.9 | 1.9 | 2.7 | 1.8 |

Table 8: Percentage of incorrect answers that are distractor values

| Prompt Size | 2k | 4k | 8k | Mean |
|---|---|---|---|---|
| Mistral-7B-v0.1 | 14.3 | 12.5 | 6.0 | 10.9 |
| StarCoder2-7B | 30.8 | 35.0 | 16.9 | 27.6 |
| StarCoderBase-1B | 15.9 | 16.2 | 22.2 | 18.1 |
| StarCoderBase-7B | 12.5 | 15.6 | 20.8 | 16.3 |
| StarCoderBase-15.5B | 19.8 | 17.9 | 15.5 | 17.7 |
| DeepSeekCoder-6.7B-base | 36.4 | 34.6 | 29.5 | 33.5 |
| GPT-4o-mini | 31.0 | 33.8 | 35.5 | 33.4 |
| GPT-4o | 25.8 | 29.1 | 24.2 | 26.3 |

Table 9: Percentage of incorrect responses in the concatenation task where the response is one of the two strings that should have been concatenated.

| Prompt Size | 2k | 4k | 8k | Mean |
|---|---|---|---|---|
| Mistral-7B-v0.1 | 20.7 | 26.7 | 11.5 | 19.6 |
| StarCoder2-7B | 21.8 | 33.5 | 30.0 | 28.5 |
| StarCoderBase-1B | 44.9 | 43.0 | 44.2 | 44.0 |
| StarCoderBase-7B | 40.9 | 50.9 | 51.6 | 47.8 |
| StarCoderBase-15.5B | 30.3 | 33.1 | 32.5 | 32.0 |
| DeepSeekCoder-6.7B-base | 23.4 | 30.4 | 34.7 | 29.5 |
| GPT-4o-mini | 28.1 | 26.7 | 32.3 | 29.0 |
| GPT-4o | 13.7 | 7.6 | 8.1 | 9.8 |

Table 10: Levenshtein Distance for incorrect responses.

| Prompt Size | 2k | 4k | 8k | Mean |
|---|---|---|---|---|
| Mistral-7B-v0.1 | 10.8 | 12.3 | 15.1 | 12.7 |
| StarCoder2-7B | 10.5 | 10.5 | 11.4 | 10.8 |
| StarCoderBase-1B | 9.6 | 9.4 | 9.7 | 9.6 |
| StarCoderBase-7B | 9.5 | 9.7 | 9.7 | 9.6 |
| StarCoderBase-15.5B | 11.1 | 11.9 | 12.3 | 11.8 |
| DeepSeekCoder-6.7B-base | 9.3 | 9.9 | 10.2 | 9.8 |
| GPT-4o-mini | 8.5 | 10.0 | 12.1 | 10.2 |
| GPT-4o | 9.5 | 11.4 | 12.4 | 11.1 |

## C Detailed Results: Call Graph Comment Experiment

We include details of model performance (accuracy@3 percent scores) on the call graph experiment.

Table 11: Call graph comment performance by model for **full-sentence template**.

| Task | Model | Comment Type | | | |
|---|---|---|---|---|---|
| | | None | Calls | Called-By | Both |
| Three Step | Mistral-7B-v0.1 | 18.2 | 16.7 | 33.8 | 39.6 |
| | StarCoder2-7B | 41.6 | 57.4 | 80.2 | 90.8 |
| | StarCoderBase-1B | 25.0 | 30.5 | 46.5 | 47.4 |
| | StarCoderBase-7B | 39.0 | 48.7 | 86.0 | 87.2 |
| | StarCoderBase-15.5B | 36.9 | 49.5 | 85.3 | 87.5 |
| | DeepSeekCoder-6.7B-base | 58.5 | 72.6 | 86.5 | 88.2 |
| | GPT-4o-mini | 64.8 | 74.2 | 82.7 | 85.4 |
| | GPT-4o | 91.6 | 95.0 | 99.0 | 97.8 |
| Concatenation | Mistral-7B-v0.1 | 2.7 | 3.8 | 5.6 | 8.9 |
| | StarCoder2-7B | 6.6 | 14.3 | 27.4 | 31.2 |
| | StarCoderBase-1B | 0.3 | 0.3 | 0.3 | 0.4 |
| | StarCoderBase-7B | 3.7 | 17.6 | 16.0 | 30.2 |
| | StarCoderBase-15.5B | 9.3 | 20.0 | 33.7 | 40.6 |
| | DeepSeekCoder-6.7B-base | 14.7 | 17.9 | 27.2 | 35.8 |
| | GPT-4o-mini | 54.1 | 65.1 | 71.1 | 73.3 |
| | GPT-4o | 95.0 | 95.3 | 96.1 | 94.0 |

Table 12: Call graph comment performance by model for **names only template**.

| Task | Model | Comment Type | | | |
|---|---|---|---|---|---|
| | | None | Calls | Called-By | Both |
| three-step | Mistral-7B-v0.1 | 18.2 | 15.0 | 36.1 | 45.4 |
| | StarCoder2-7B | 41.6 | 47.5 | 72.0 | 71.8 |
| | StarCoderBase-1B | 25.0 | 28.9 | 51.5 | 48.6 |
| | StarCoderBase-7B | 39.0 | 35.5 | 80.1 | 71.6 |
| | StarCoderBase-15.5B | 36.9 | 35.5 | 80.8 | 63.0 |
| | DeepSeekCoder-6.7B-base | 58.5 | 60.0 | 83.6 | 80.5 |
| | GPT-4o-mini | 64.8 | 66.6 | 81.4 | 78.5 |
| | GPT-4o | 91.6 | 93.1 | 97.9 | 94.9 |
| concatenation | Mistral-7B-v0.1 | 2.7 | 2.3 | 4.0 | 4.1 |
| | StarCoder2-7B | 6.6 | 8.5 | 15.4 | 11.3 |
| | StarCoderBase-1B | 0.3 | 0.2 | 0.4 | 0.3 |
| | StarCoderBase-7B | 3.7 | 6.9 | 4.0 | 11.4 |
| | StarCoderBase-15.5B | 9.3 | 11.2 | 24.6 | 19.2 |
| | DeepSeekCoder-6.7B-base | 14.7 | 15.2 | 23.4 | 23.8 |
| | GPT-4o-mini | 54.1 | 58.2 | 68.9 | 70.5 |
| | GPT-4o | 95.0 | 94.6 | 94.9 | 95.7 |

Table 13: Next-hop vs. full graph comments performance on three-step retrieval.

| Model | # Forward References | Comment Type | | |
|---|---|---|---|---|
| | | None | Next Hop | Full Graph |
| Mistral-7B-v0.1 | 0 | 16.8 | 39.6 | 53.4 |
| | 1 | 19.5 | 29.9 | 39.8 |
| | 2 | 14.7 | 17.1 | 25.3 |
| StarCoder2-7B | 0 | 40.7 | 83.2 | 90.9 |
| | 1 | 43.9 | 80.5 | 92.5 |
| | 2 | 33.1 | 61.1 | 84.3 |
| StarCoderBase-1B | 0 | 27.2 | 28.4 | 32.9 |
| | 1 | 27.3 | 46.3 | 52.1 |
| | 2 | 13.8 | 29.9 | 43.4 |
| StarCoderBase-7B | 0 | 56.9 | 88.7 | 92.5 |
| | 1 | 39.6 | 78.9 | 91.2 |
| | 2 | 18.4 | 46.8 | 66.0 |
| StarCoderBase-15.5B | 0 | 48.6 | 66.6 | 76.6 |
| | 1 | 37.1 | 78.7 | 89.3 |
| | 2 | 24.2 | 80.2 | 91.5 |
| DeepSeekCoder-6.7B-base | 0 | 54.8 | 84.5 | 87.9 |
| | 1 | 61.7 | 86.1 | 91.0 |
| | 2 | 49.6 | 66.7 | 77.7 |
| GPT-4o-mini | 0 | 53.6 | 70.0 | 75.2 |
| | 1 | 68.4 | 84.3 | 88.1 |
| | 2 | 61.4 | 78.7 | 84.7 |
| GPT-4o | 0 | 89.8 | 98.1 | 97.8 |
| | 1 | 93.0 | 98.3 | 97.9 |
| | 2 | 87.6 | 96.9 | 97.7 |

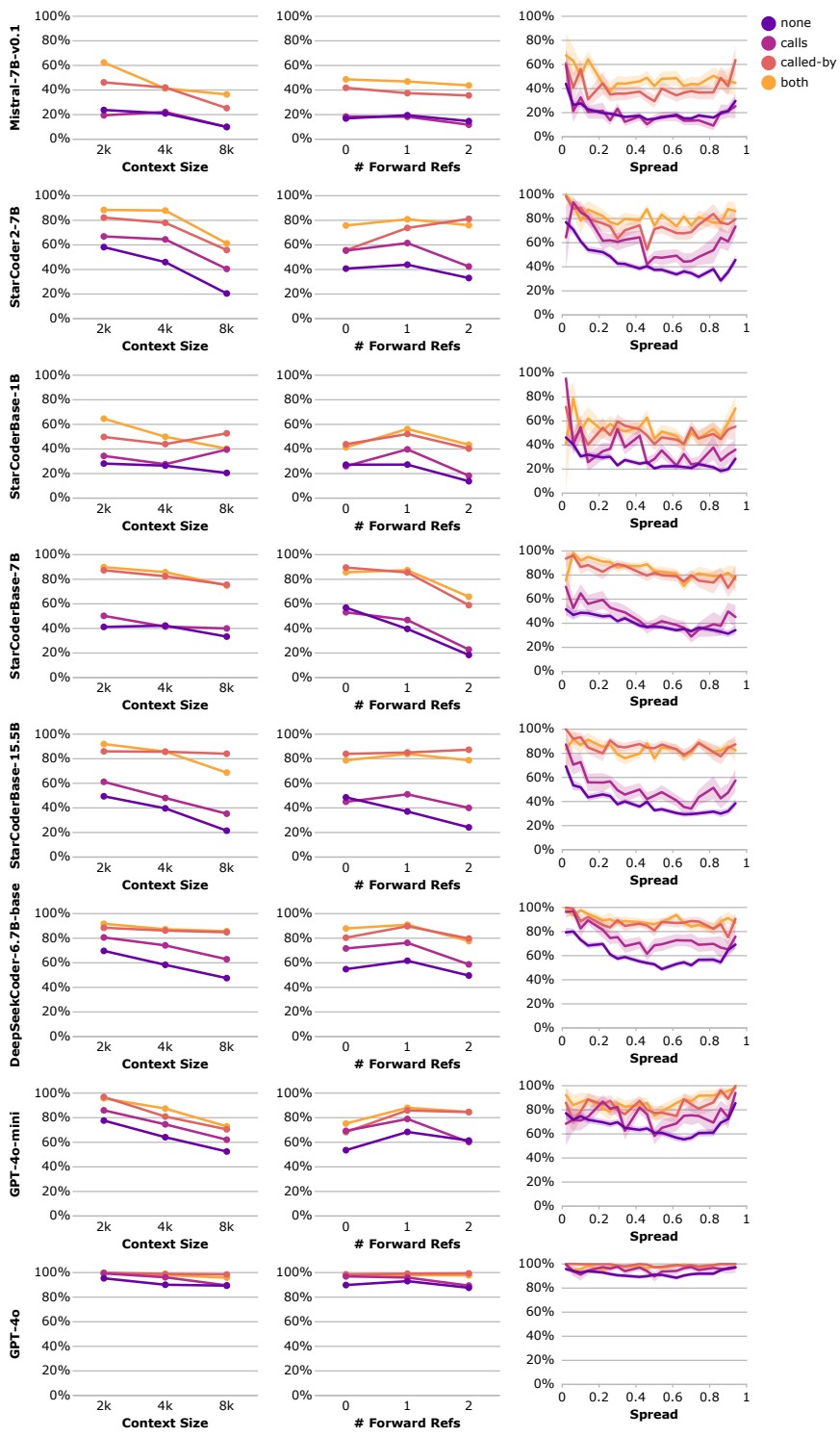

Figure 15: Effect of call graph comments for **three-step** task.

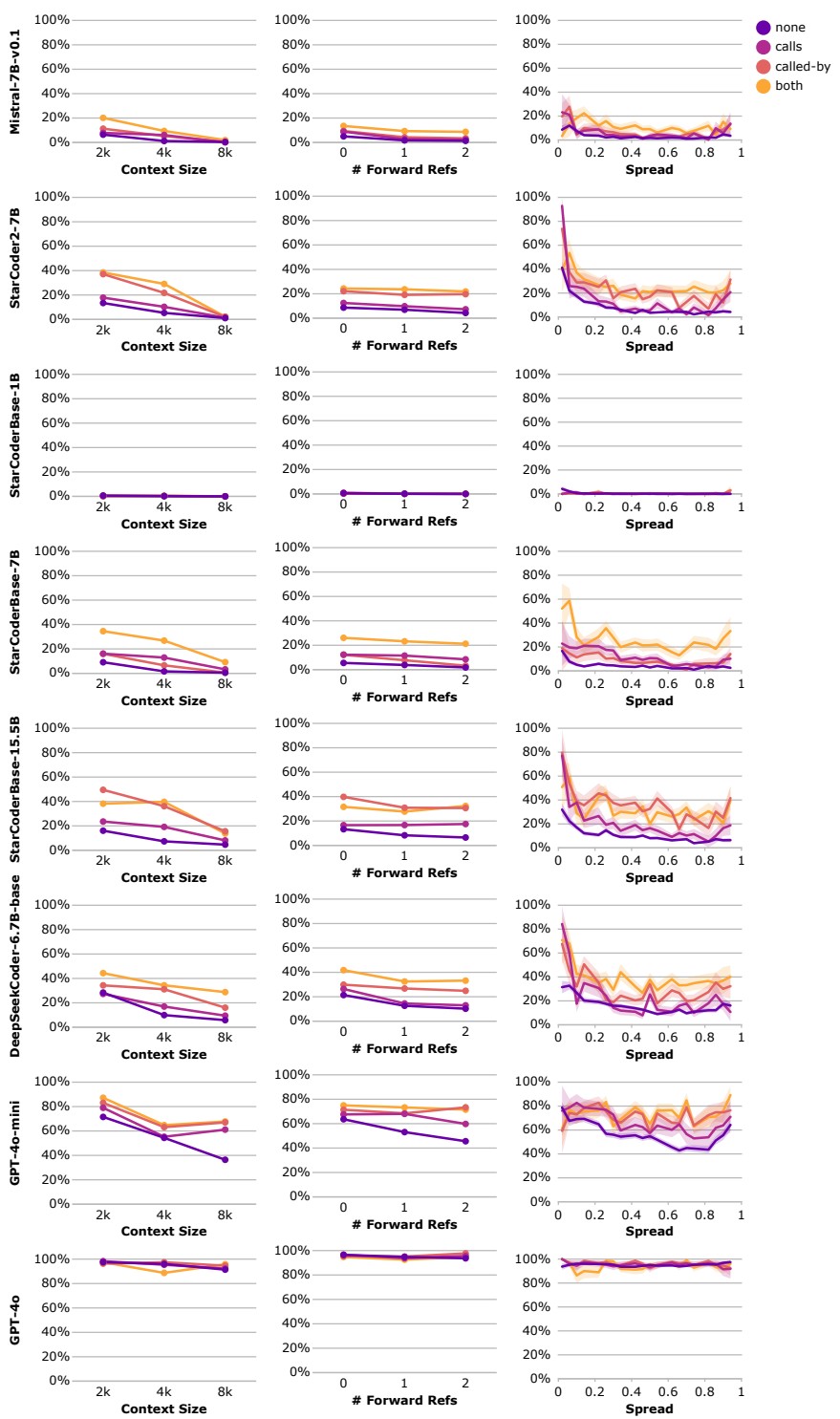

Figure 16: Effect of call graph comments for **concatenation** task.

## D   Hyperparameters for Generation

For all experiments we use nucleus sampling with the following parameters.

Table 14: Generation hyperparameters.

| Hyperparameter | Value |
|---|---|
| Temperature | 0.8 |
| Top p | 0.95 |
| Top k | 0 |
| Batch size | 1 |
| Output samples per input prompt | 10 |

## E   Confidence Intervals

All confidence intervals (error bars and plus/minus values) are calculated as 95% confidence intervals of the estimated mean following assumptions of normal distribution and independent samples.

## F   Compute Information

We ran all experiments on machines with a single A100 GPU with 80GB of VRAM on a cloud provider. Experiments were run over the course of about a month and we estimate ~1000 GPU hours were used including runs that had bugs in them and additional experiments that are not part of the paper. We use FlashAttention Dao et al. (2022) to improve memory usage and latency of generation.

## G   Prompt Samples for Multi Step Key Retrieval

```python
def iaizjb_184360_440195():
    return "wxmbrnpokw"

def string_to_md5(text):
    """
    Given a string 'text', return its md5 hash equivalent string.
    If 'text' is an empty string, return None.

    >>> string_to_md5('Hello world') == '3e25960a79dbc69b674cd4ec67a72c62'
    """
    import hashlib
    return hashlib.md5(text.encode('ascii')).hexdigest() if text else None

def egllun_467846():
    return "eooyfwmxln"

def count_upper(s):
    """
    Given a string s, count the number of uppercase vowels in even indices.

    For example:
    count_upper('aBCdEf') returns 1
    count_upper('abcdefg') returns 0
    count_upper('dBBE') returns 0
    """
    count = 0
    for i in range(0,len(s),2):
        if s[i] in "AEIOU":
            count += 1
    return count

def solution(lst):
    """Given a non-empty list of integers, return the sum of all of the odd elements that are in even positions
    .

    Examples
    solution([5, 8, 7, 1]) ==> 12
    solution([3, 3, 3, 3, 3]) ==> 9
    solution([30, 13, 24, 321]) ==>0
    """
    return sum([x for idx, x in enumerate(lst) if idx%2==0 and x%2==1])

def choose_num(x, y):
    """This function takes two positive numbers x and y and returns the
    biggest even integer number that is in the range [x, y] inclusive. If
    there's no such number, then the function should return -1.

    For example:
    choose_num(12, 15) = 14
    choose_num(13, 12) = -1
    """
    if x > y:
        return -1
    if y % 2 == 0:
        return y
    if x == y:
        return -1
    return y - 1

def digitSum(s):
    """Task
    Write a function that takes a string as input and returns the sum of the upper characters only'
    ASCII codes.

    Examples:
        digitSum("") => 0
        digitSum("abAB") => 131
        digitSum("abcCd") => 67
        digitSum("helloE") => 69
        digitSum("woArBld") => 131
        digitSum("aAaaaXa") => 153
    """
    if s == "": return 0
    return sum(ord(char) if char.isupper() else 0 for char in s)

def multiply(a, b):
    """Complete the function that takes two integers and returns
```

```
    the product of their unit digits.
    Assume the input is always valid.
    Examples:
    multiply(148, 412) should return 16.
    multiply(19, 28) should return 72.
    multiply(2020, 1851) should return 0.
    multiply(14,-15) should return 20.
    """
    return abs(a % 10) * abs(b % 10)

def right_angle_triangle(a, b, c):
    '''
    Given the lengths of the three sides of a triangle. Return True if the three
    sides form a right-angled triangle, False otherwise.
    A right-angled triangle is a triangle in which one angle is right angle or
    90 degree.
    Example:
    right_angle_triangle(3, 4, 5) == True
    right_angle_triangle(1, 2, 3) == False
    '''
    return a*a == b*b + c*c or b*b == a*a + c*c or c*c == a*a + b*b

def add_elements(arr, k):
    """
    Given a non-empty array of integers arr and an integer k, return
    the sum of the elements with at most two digits from the first k elements of arr.

    Example:

        Input: arr = [111,21,3,4000,5,6,7,8,9], k = 4
        Output: 24 # sum of 21 + 3

    Constraints:
        1. 1 <= len(arr) <= 100
        2. 1 <= k <= len(arr)
    """
    return sum(elem for elem in arr[:k] if len(str(elem)) <= 2)

def vskfby_510934():
    return "thwtyqwjws"

def qgtsin_336194_iwdghb():
    return iaizjb_184360_440195()

def awdpgq_293061_vwetvu():
    return rbwofb_803321_331141()

def even_odd_count(num):
    """Given an integer. return a tuple that has the number of even and odd digits respectively.

     Example:
        even_odd_count(-12) ==> (1, 1)
        even_odd_count(123) ==> (1, 2)
    """
    even_count = 0
    odd_count = 0
    for i in str(abs(num)):
        if int(i)%2==0:
            even_count +=1
        else:
            odd_count +=1
    return (even_count, odd_count)

def minSubArraySum(nums):
    """
    Given an array of integers nums, find the minimum sum of any non-empty sub-array
    of nums.
    Example
    minSubArraySum([2, 3, 4, 1, 2, 4]) == 1
    minSubArraySum([-1, -2, -3]) == -6
    """
    max_sum = 0
    s = 0
    for num in nums:
        s += -num
        if (s < 0):
            s = 0
        max_sum = max(s, max_sum)
    if max_sum == 0:
        max_sum = max(-i for i in nums)
    min_sum = -max_sum
```

```python
    return min_sum

def qvkrlp_701464_787942():
    return vskfby_510934()

def rdcxoi_135343():
    return egllun_467846()

def iscube(a):
    '''
    Write a function that takes an integer a and returns True
    if this ingeger is a cube of some integer number.
    Note: you may assume the input is always valid.
    Examples:
    iscube(1) ==> True
    iscube(2) ==> False
    iscube(-1) ==> True
    iscube(64) ==> True
    iscube(0) ==> True
    iscube(180) ==> False
    '''
    a = abs(a)
    return int(round(a ** (1. / 3))) ** 3 == a

def special_factorial(n):
    """The Brazilian factorial is defined as:
    brazilian_factorial(n) = n! * (n-1)! * (n-2)! * ... * 1!
    where n > 0

    For example:
    >>> special_factorial(4)
    288

    The function will receive an integer as input and should return the special
    factorial of this integer.
    """
    fact_i = 1
    special_fact = 1
    for i in range(1, n+1):
        fact_i *= i
        special_fact *= fact_i
    return special_fact

def is_equal_to_sum_even(n):
    """Evaluate whether the given number n can be written as the sum of exactly 4 positive even numbers
    Example
    is_equal_to_sum_even(4) == False
    is_equal_to_sum_even(6) == False
    is_equal_to_sum_even(8) == True
    """
    return n%2 == 0 and n >= 8

assert rdcxoi_135343() ==
```

Listing 1: Sample prompt for two-step retrieval task with 2000 token context size and 5 distractors. **rdcxoi__135343** is the key function and **egllun__467846** is the value function.

## H    Prompt Samples for Call Graph Comments Experiment

```
# This function calls oswgtr_325169_862229
# This function is called by aokwfl_208971_hwmofh
def gqpvbp_138573():
    return oswgtr_325169_862229()

# This function calls gqpvbp_138573 and oswgtr_325169_862229
def aokwfl_208971_hwmofh():
    return gqpvbp_138573()

# This function calls lhezee_508969 and hjdnwl_724283
def oftoyy_286138():
    return lhezee_508969()

# This function is called by gqpvbp_138573 and aokwfl_208971_hwmofh
def oswgtr_325169_862229():
    return "kyfgholcrg"

# This function calls gjobme_651008_tymmij
# This function is called by wwzfoa_904885
def bweckw_860527_nykiyp():
    return gjobme_651008_tymmij()

# This function is called by lhezee_508969 and oftoyy_286138
def hjdnwl_724283():
    return "pwincnzyqh"

# This function is called by bweckw_860527_nykiyp and wwzfoa_904885
def gjobme_651008_tymmij():
    return "axxtrhucug"

# This function calls bweckw_860527_nykiyp and gjobme_651008_tymmij
def wwzfoa_904885():
    return bweckw_860527_nykiyp()

assert wwzfoa_904885() ==
```

Listing 2:  Full sentence call graph comments for three step retrieval.  5 distractors.  HumanEval functions excluded for brevity. **wwzfoa__904885** is the key function, **bweckw__860527__nykiyp** and **gjobme__651008__tymmij** are value functions

```python
# oswgtr_325169_862229
# aokwfl_208971_hwmofh
def gqpvbp_138573():
    return oswgtr_325169_862229()

# gqpvbp_138573, oswgtr_325169_862229
def aokwfl_208971_hwmofh():
    return gqpvbp_138573()

# lhezee_508969, hjdnwl_724283
def oftoyy_286138():
    return lhezee_508969()

# gqpvbp_138573, aokwfl_208971_hwmofh
def oswgtr_325169_862229():
    return "kyfgholcrg"

# gjobme_651008_tymmij
# wwzfoa_904885
def bweckw_860527_nykiyp():
    return gjobme_651008_tymmij()

# lhezee_508969, oftoyy_286138
def hjdnwl_724283():
    return "pwincnzyqh"

# bweckw_860527_nykiyp, wwzfoa_904885
def gjobme_651008_tymmij():
    return "axxtrhucug"

# bweckw_860527_nykiyp, gjobme_651008_tymmij
def wwzfoa_904885():
    return bweckw_860527_nykiyp()
```

Listing 3: Function names only call graph comments for three step retrieval. 5 distractors. HumanEval functions excluded for brevity. **wwzfoa_904885** is the key function, **bweckw_860527_nykiyp** and **gjobme_651008_tymmij** are value functions

