# OpenReview forum: "Evaluating Long Range Dependency Handling in Code Generation LLMs"
_TMLR — Accepted by TMLR_

### Review · Reviewer_txvE · 2025-03-02

**Summary Of Contributions:**

The paper focuses on code generation models and how well they handle dependencies spread throughout a long context. Instead of just testing whether models can find a single piece of information (like the popular "needle-in-a-haystack" tests), they create more challenging tasks that require models to follow a chain of function calls to arrive at the correct answer.
Their experiments reveal some surprising limitations. Most notably, models struggle significantly when they encounter functions that reference other functions defined later in the prompt, a common pattern in real codebases. Models using sliding window attention also hit a performance cliff when dependencies stretch beyond their window size, even though they technically support much larger contexts.
Perhaps most valuable is their discovery that a simple fix, adding comments that describe function relationships, can dramatically improve performance. This provides an immediately useful technique for anyone working with these models.

Contributions:
- The evaluation framework to properly study where models start to struggle with long-range dependencies.
- They showed weaknesses in how current models handle code, particularly with forward references and dependencies that span across large distances in the context window.
- They provided a practical solution: adding call graph annotations (that significantly improves performance without requiring any model retraining or complex processing).

**Audience:**

Yes

**Claims And Evidence:**

Yes

**Requested Changes:**

I would suggest to change the structure of the paper:

The authors have this fascinating insight about forward references causing problems, but we don't get a clear theoretical explanation for why this might be happening. Is it related to how attention works in these models? Is it because training data typically follows certain patterns? These questions deserve exploration before jumping into experiment details.

The solution of adding call graph comments is genuinely clever and practical, but it feels like it comes out without a theoretical setup first: drawing connections to the chain-of-thought work by Wei and Kojima mentioned in the related work section would help readers understand why this approach makes sense theoretically.

The experiments themselves are well-designed and thorough, especially those progressive difficulty tasks they created. I just wish I had understood the conceptual framework behind them earlier in my reading journey.


I'd say this is the only change i'd suggest, and it's a suggestion to strengthen the paper, not a condition for acceptace. I really think though that changing the structure would make the paper look more robust.

**Strengths And Weaknesses:**

Weaknesses:

1) Limited exploration of prompt position effects:
while the study examines snippet position effects, it doesn't fully explore whether placing the assert statement (the task instruction) at different positions within the context might give different results.

2) Missing analysis of attention patterns:
the paper could benefit from an analysis of attention patterns to better understand the mechanisms behind the observed performance differences.

3) Missing analysis of other possible settings:
while the paper suggests prompt-based improvements, it doesn't explore retrieval augmentation (many long-context applications use retrieval augmentation), fine-tuning, a few shot setting, or cross-modality testing.



Strengths:

1) The experiments settings and design:
- Comprehensive model evaluation: the paper study a diverse set of models, including both open-source (StarCoder family, Mistral, DeepSeekCoder) and proprietary models (GPT-4o variants).
- Reproducibility;
- Clear visualization of results;

2) The evaluation methodology: the paper goes further the simple "needle-in-a-haystack" retrieval tasks by requiring models to reason through function calls and integrate multiple pieces of information spread throughout the context.

3) The paper doesn't just identify problems but propose and test practical solutions (and the call graph annotation solution they propose requires no model retraining and minimal computational overhead)

---

> ### Author Response · Authors · 2025-03-28
> **Response to reviewer txvE Part 1**
>
> We thank the reviewer for their thoughtful and encouraging feedback. We're glad they found the experiments well-designed and the overall evaluation approach thorough. We appreciate the suggestions for improving the conceptual framing and situating our findings in a broader methodological context, which we believe will help sharpen the paper.
>
> >Limited exploration of prompt position effects: while the study examines snippet position effects, it doesn't fully explore whether placing the assert statement (the task instruction) at different positions within the context might give different results.
>
>
> We thank the reviewer for this observation. Keeping the `assert` statement at the end allowed us to use a consistent prompt construction method across models, including those that do not support fill-in-the-middle (e.g., GPT-4o). For models that do support that functionality, exploring whether the position of the completion target affects performance would be an interesting direction for future work.
>
>
> >The authors have this fascinating insight about forward references causing problems, but we don't get a clear theoretical explanation for why this might be happening. Is it related to how attention works in these models? Is it because training data typically follows certain patterns? These questions deserve exploration before jumping into experiment details.
>
>
> We agree that these are important questions and acknowledge that our paper does not fully explore the underlying mechanisms. We had also hypothesized in the paper that causal attention could contribute to the observed behavior, and agree that patterns in the training data might play a role. While we felt a detailed analysis was beyond the scope of this paper, we see multiple promising directions for future work.
>
> As the reviewer noted, attention analysis may help uncover representational differences between forward and non-forward references—though it remains debated whether attention patterns are reliable explanations [1,2].
>
> Beyond attention, emerging work in mechanistic interpretability offers tools that may be well-suited to studying behaviors like forward reference handling. For example, Elhage et al. [3] identify "induction heads" in small transformers responsible for copying earlier tokens based on matching prefixes. Hernandez et al. [4] investigate how transformers represent relations between entities, and Brinkmann et al. [5] analyze models solving symbolic multi-step reasoning tasks (e.g., finding paths in binary trees). These approaches aim to isolate the circuits and mechanisms responsible for specific behaviors.
>
> While such techniques are not yet widely scaled to long contexts or reasoning over distant tokens, we believe our benchmark could provide a useful testbed for further development—particularly in settings where function ordering or reference directionality can be manipulated in controlled ways.
>
> We also see the potential value in training models under controlled conditions—e.g., comparing causal vs. bidirectional attention, or varying the distribution of forward references in the training data—as a way to test these hypotheses about the mechanisms underlying this behavior.
>
> We will update our discussion to draw a connection to this literature in considering paths forward for more in-depth understanding of these phenomena.
>
> [1] Attention is Not Explanation, Jain & Wallace 2019 (https://arxiv.org/pdf/1902.10186)
>
> [2] Attention is Not Not Explanation, Weigreffe, Pinter 2019 (https://arxiv.org/abs/1908.04626)
>
> [3] A Mathematical Framework for Transformer Circuits, Elhage et al 2021 (https://transformer-circuits.pub/2021/framework/index.html)
>
> [4] Linearity of Relation Decoding in Transformer LMs, Hernandez et al 2024 (https://arxiv.org/abs/2308.09124)
>
> [5] A Mechanistic Analysis of a Transformer Trained on a Symbolic Multi-Step Reasoning Task, Brinkmann et al, 2024 (https://arxiv.org/abs/2402.11917)
>
> **Response continues below**

---

> > ### Author Response · Authors · 2025-03-28
> > **Response to reviewer txvE Part 2**
> >
> > >Missing analysis of other possible settings: while the paper suggests prompt-based improvements, it doesn't explore retrieval augmentation (many long-context applications use retrieval augmentation), fine-tuning, a few shot setting, or cross-modality testing.
> >
> >
> > We agree that our experiments do not evaluate this intervention against other paradigms such as fine-tuning, retrieval augmentation, or few-shot prompting. Our primary motivation for introducing the call graph comments was to better understand the forward reference phenomenon. We treated them as a lightweight mechanism to reintroduce the relevant function tokens earlier in the prompt—without changing the code order—so as to test whether token-level presence could mitigate performance degradation.
> >
> > We do not claim that this approach outperforms other methods, but found it surprising how much improvement it yielded given its simplicity. We will clarify this in the limitations section. We do believe these findings suggest interesting questions for more complex setups. For example, in retrieval-augmented systems, the placement of retrieved snippets could interact with observed behaviors. While beyond our current scope, we hope our results provide useful evidence to inform such future investigations.
> >
> >
> > >The experiments themselves are well-designed and thorough, especially those progressive difficulty tasks they created. I just wish I had understood the conceptual framework behind them earlier in my reading journey.
> >
> >
> >
> > We appreciate this comment and agree that clarifying the motivation and conceptual framing earlier would strengthen the paper. To improve this, we plan the following changes:
> >
> > * Call graph experiments
> >     * Expand the introduction to Section 4 to more clearly connect the intervention to the forward reference findings and explain the initial motivating rationale—namely, that introducing tokens corresponding to called functions might offset ordering effects.
> >     * Draw a clearer conceptual link to chain-of-thought prompting (e.g., Wei et al.) and reasoning-focused prompting strategies, where additional structure helps with multi-step reasoning.
> > * Discussion
> >     * Situate our findings more explicitly in relation to work on how transformers encode and transmit information across tokens.
> > * Limitations
> >     * Add a caveat that our annotation-based intervention is exploratory and may not generalize across settings or outperform stronger alternatives such as fine-tuning or prompt tuning.

---

### Review · Reviewer_dwJc · 2025-03-04

**Summary Of Contributions:**

The authors evaluate long-range dependency handling in code generation. The authors introduce various tasks with increasing complexity to assess how models use information spread within a prompt. Their experiments with both open source and proprietary models reveal that performance degrades significantly when functions reference definitions that appear later in the prompt. They also find that models using sliding window attention struggle with dependencies beyond a single window size. They also demonstrate that adding call graph information via simple comments can improve multi-step retrieval performance.

**Audience:**

Yes

**Broader Impact Concerns:**

No notable ethical considerations

**Claims And Evidence:**

Yes

**Requested Changes:**

Content:

- Improve the discussion by placing the work in context of related literature, and more clearly state key takeaways
- Add to the limitations, both in depth and breadth
- Is there any sense for how much "prefix_allowed_tokens" affects general performance of the models? Can we trust this to be negligible? (See also weakness)

Methods:

- End of section three talks about three distractor conditions but I believe there are five. Also, a better clarification between the condition vs the distractor function would be helpful
- Below Algorithm 1 is a list of the variables to be studied, n_p is not listed, should it be?
- In the prompt samples in the appendix it would be good to get a description in the caption of what the target and intermediate functions are

Figures and Tables:

 - The caption for figures 2 and 3 don't mention this is mean for 10 runs (I think this would be helpful information), also it would be very helpful to state that the variance is minimal and error bars would not be visible – to save the reader from wondering what the variance is and having to look at the appendix tables
- Describe the shaded region in Figure 4 in the caption
- The colours in the colourbars for figure 6 need to be fixed
- Adding the names of the models in Figure 7 (perhaps on the left side) would make it more readable
- There are no variances in table 2, were these single runs?
- The statistic used to describe the distributions in appendix table 5 needs to be described (standard dev, standard error, etc)

Formatting/Text:

- The footnotes in table 1 on page 5 actually appear on page 4, I believe they should be on the same page
- There is a strange gap in the opening line on page 6, it matches with the inset figure on the page before but looks strange on page 7
- Start of section 3.5.3 "five distractor condition" → "conditions"

**Strengths And Weaknesses:**

Strengths
- Fills a research gap with a balance between task realism and control
- It is well-written, structured, and easy to follow. In most places a good-quality manuscript
- The experiments are suitable and provide decent coverage of models, although Llama is a notable omission
- It will be of value to the community and particularly TMLR readers

Weaknesses
- The discussion is a weak point, lacking depth particularly in placing the work in context of the related literature, which is completely missing. The related work (section 2) is good, but it would tie the paper together better to refer back to previous findings. In particular, it would be valuable to see this discussed both with respect to Zhang et al. (2022) and Yu et al. (2024)
- The limitations section is also quite weak. A deeper explanation of the key differences between the synthetic code and real code, along with details on how this might affect the results, would improve it. There are also many limitations not discussed (lack of variety in code tests, realism, models tested, inability to test different context windows for all models), and some of those have clear justifications which should make it easy to detail
- The GPT4o models were run without the "allowed tokens" filter, which puts it at a disadvantage due to a technical limitation (API not supported). However, it would be better to get a sense of the impact of this, either by checking how often GPT4o produces a trivial solution or the performance of other models with this feature turned off (putting them on par with GPT4o). Even an estimate of the order of magnitude of these differences would be helpful
- It would have been stronger with more models, particularly some llama model (even though they are general purpose models), however this current set is sufficient to demonstrate the claims made
- There are some more minor issues that can be improved, which I will list below in requested changes

---

> ### Author Response · Authors · 2025-03-28
> **Response to reviewer dwJc Part 1**
>
> We thank the reviewer for their helpful and detailed feedback and for noting the gap this work fills in the literature. We appreciate the emphasis on improving the contextualization of our work and expanding the discussion of limitations. We respond to individual points below.
>
>
> >The discussion is a weak point, lacking depth particularly in placing the work in context of the related literature, which is completely missing. The related work (section 2) is good, but it would tie the paper together better to refer back to previous findings. In particular, it would be valuable to see this discussed both with respect to Zhang et al. (2022) and Yu et al. (2024)
>
>
> We agree that connecting our findings back to prior work would strengthen the discussion and thank the reviewer for these suggestions. We plan to incorporate the following into our discussion:
>
> * Our findings align with those of Yu et al. (2024), who show that over 70% of functions in popular open-source repositories are “non-standalone” and report a 48% average performance drop when models are evaluated on such functions. This underscores the importance of evaluating the ability of models to reason over interdependent code units. Our work complements theirs by using synthetic code to expand the context beyond the 1k token limit in their benchmark.
> * Zhang et al. (2022) introduced Pointer Value Retrieval (PVR) tasks and describe challenges neural models have with learning to handle indirection. Follow-up work by Abnar et al. (2023) extended this to multi-hop settings and exploring adaptive architectures to better address such challenges. Inspired by these efforts, we design multi-step retrieval tasks similarly centered on indirection, but specifically targeting evaluating inference-time limitations of pre-trained code models, and expand the input design space to include the effects of position and ordering, in addition to the number of hops.
> * Our findings shed light on specific factors influencing the fragility of multi-step reasoning in large language models, including the presence of distractors, function ordering and architectural constraints such as sliding-window attention.
>
> We believe these suggested changes can help strengthen the presentation of the work, and hope we have captured the reviewer's suggestion.
>
>
> >Add to the limitations, both in depth and breadth
>
> >There are also many limitations not discussed (lack of variety in code tests, realism, models tested, inability to test different context windows for all models), and some of those have clear justifications which should make it easy to detail
>
>
> We appreciate this feedback and will expand the limitations section to include the following points (as well as a couple raised by reviewer txvE):
>
> * **Limited code patterns**: Our tasks focus on top-level functions composed via call chains and concatenation. We do not test more complex structures of indirection such as recursion, branching control flow, or shared abstractions that cause functions in the call graph to be reused (e.g. helper functions used by multiple functions). These may introduce additional retrieval challenges or sensitivity to ordering not captured by our current setup.
> * **Out-of-domain naming/Realism**: Our use of randomly generated function names results in a highly out-of-domain setting. Models may perform better with more realistic identifiers drawn from natural codebases.
> * **Architectural scope**: We evaluate only decoder-only causal transformers. Models with recurrence or bidirectional attention may behave differently and merit separate study.
>
>
> We may have misunderstood the reviewer’s point about “inability to test different context windows”. All models were evaluated across the 3 context windows sizes in our experiments (2k, 4k, 8k). For the sliding window experiments; only Mistral allowed us to modify the sliding window size at runtime (and only one other model uses sliding window). If we’ve misunderstood the concern, we’d be happy to clarify further.
>
> >End of section three talks about three distractor conditions but I believe there are five. Also, a better clarification between the condition vs the distractor function would be helpful
>
> We do see how the wording may have caused confusion. There are three *distractor conditions*—zero, one, and five distractors (number of distractor functions in that condition). We’ll clarify this language and use hyphenation where appropriate (e.g., “5-distractor condition”) to distinguish between the number of distractors and number of conditions.
>
> **Response continues below**

---

> ### Author Response · Authors · 2025-03-28
> **Response to reviewer dwJc Part 2**
>
> >The GPT4o models were run without the "allowed tokens" filter, which puts it at a disadvantage due to a technical limitation (API not supported) ... How often does GPT4o produce a trivial solution
>
>
> We note that in order to compensate for the lack of guided decoding, we slightly modified the prompts passed to the GPT models to end with a double quote, as a result the assert statement would be something like:
>
> ```
> assert bvncxl_229753() == \"
> ```
>
> We regret that we omitted this detail from the footnote and will add it in the revision. We ran the experiments on the other models significantly earlier than the GPT experiments and they were quite time/compute intensive and thus we did not go back to use this approach for the other models. However, we believe the modified prompt places the GPT models on comparable footing with other models.
>
> We did take a look at how often the GPT models produced the trivial solution (i.e. returning the key function itself) and observed the following rates:
>
> - GPT-4o:  0.000160%
> - GPT-4o-mini: 0.000748%
>
> We see it is quite rare and do not believe it significantly impacts the reported results.
>
>
> >There are no variances in table 2, were these single runs?
>
>
> The values are mean pass-at-3 accuracy for prompts on each condition (each prompt is run 10 per model to compute pass-at-3). We will incorporate variance information into this table in the manuscript. The variances are generally small and the maximum variance (size of the 95% confidence interval) across the table is 3.0%.
>
> Here is a preview of the updated table with confidence intervals:
>
> Column headers like 2k, 0 are “prompt length, number of forward references”. 95% confidence intervals shown in parentheses.
>
>
> ```
> # Two-step
>
> | Model                   | 2k, 0      | 2k, 1      | 4k, 0      | 4k, 1      | 8k, 0      | 8k, 1      |
> | :---------------------- | :--------- | :--------- | :--------- | :--------- | :--------- | :--------- |
> | Mistral-7B-v0.1         | 60.1 (1.1) | 21.1 (1.2) | 46.4 (1.3) | 22.5 (1.3) | 8.0 (1.3)  | 9.2 (1.3)  |
> | StarCoder2-7B           | 73.5 (0.1) | 57.6 (0.4) | 69.4 (0.4) | 56.5 (0.8) | 28.1 (0.6) | 26.8 (1.2) |
> | StarCoderBase-1B        | 55.3 (0.4) | 39.4 (0.7) | 49.7 (1.0) | 33.9 (1.1) | 65.6 (1.1) | 39.3 (1.3) |
> | StarCoderBase-7B        | 90.0 (1.4) | 38.1 (1.1) | 87.2 (1.4) | 34.1 (1.1) | 76.0 (0.8) | 37.7 (0.8) |
> | StarCoderBase-15.5B     | 85.5 (1.3) | 63.4 (1.4) | 82.3 (1.4) | 49.2 (1.4) | 63.4 (1.4) | 28.3 (1.3) |
> | DeepSeekCoder-6.7B-base | 83.7 (1.0) | 75.2 (1.3) | 74.4 (1.0) | 67.7 (1.4) | 70.2 (1.3) | 58.5 (1.3) |
> | GPT-4o-mini             | 98.4 (1.4) | 95.7 (1.4) | 91.4 (1.4) | 87.2 (1.3) | 87.4 (1.4) | 80.3 (1.4) |
> | GPT-4o                  | 99.8 (0.8) | 98.3 (1.3) | 98.6 (0.9) | 93.6 (1.3) | 96.2 (1.2) | 83.9 (1.3) |
>
> # Three-step
>
> | Model                   | 2k, 0      | 2k, 1      | 2k, 2      | 4k, 0      | 4k, 1      | 4k, 2      | 8k, 0      | 8k, 1      | 8k, 2      |
> | :---------------------- | :--------- | :--------- | :--------- | :--------- | :--------- | :--------- | :--------- | :--------- | :--------- |
> | Mistral-7B-v0.1         | 28.1 (2.1) | 24.4 (1.0) | 16.8 (2.2) | 19.8 (2.3) | 22.3 (1.1) | 16.9 (2.1) | 2.7 (2.3)  | 11.7 (1.2) | 10.3 (2.1) |
> | StarCoder2-7B           | 61.5 (1.1) | 60.5 (0.5) | 45.7 (1.8) | 46.2 (1.9) | 48.1 (0.8) | 37.5 (2.0) | 14.4 (2.0) | 23.1 (0.8) | 16.0 (1.9) |
> | StarCoderBase-1B        | 32.3 (2.8) | 29.5 (1.1) | 18.1 (2.7) | 29.5 (3.0) | 28.5 (1.4) | 14.9 (2.8) | 19.8 (2.8) | 23.8 (1.4) | 8.2 (2.8)  |
> | StarCoderBase-7B        | 63.4 (2.0) | 41.3 (0.9) | 18.8 (1.7) | 65.4 (1.8) | 42.0 (0.9) | 20.5 (1.6) | 41.9 (0.8) | 35.6 (0.8) | 15.9 (1.3) |
> | StarCoderBase-15.5B     | 57.2 (2.4) | 50.6 (1.2) | 37.2 (2.4) | 54.0 (2.4) | 39.7 (1.3) | 24.9 (2.4) | 34.5 (1.8) | 21.0 (1.1) | 10.5 (1.7) |
> | DeepSeekCoder-6.7B-base | 69.4 (2.4) | 73.1 (1.2) | 56.2 (2.3) | 52.2 (2.3) | 61.3 (1.2) | 52.6 (1.9) | 42.9 (2.2) | 50.5 (1.0) | 40.1 (1.3) |
> | GPT-4o-mini             | 68.0 (2.2) | 81.8 (1.1) | 70.9 (1.9) | 53.3 (2.3) | 67.1 (1.1) | 62.8 (1.8) | 39.4 (2.0) | 56.3 (1.1) | 50.4 (1.3) |
> | GPT-4o                  | 95.7 (2.2) | 96.7 (1.2) | 89.6 (1.6) | 88.7 (2.0) | 91.3 (1.1) | 86.6 (1.8) | 85.2 (2.3) | 91.0 (1.1) | 86.8 (1.6) |
>
> **Response continues below**

---

> ### Author Response · Authors · 2025-03-28
> **Response to reviewer dwJc Part 3**
>
> ```
> # Concatenation
>
> | Model                   | 2k, 0      | 2k, 1      | 2k, 2      | 4k, 0      | 4k, 1      | 4k, 2      | 8k, 0      | 8k, 1      | 8k, 2      |
> | :---------------------- | :--------- | :--------- | :--------- | :--------- | :--------- | :--------- | :--------- | :--------- | :--------- |
> | Mistral-7B-v0.1         | 11.8 (1.8) | 4.3 (1.5)  | 3.1 (1.3)  | 2.4 (1.2)  | 0.6 (0.9)  | 0.6 (0.8)  | 0.8 (1.0)  | 0.2 (0.6)  | 0.2 (0.5)  |
> | StarCoder2-7B           | 17.7 (0.4) | 14.7 (0.6) | 7.9 (0.6)  | 6.4 (0.6)  | 5.4 (0.7)  | 4.2 (0.8)  | 1.8 (0.8)  | 0.7 (0.9)  | 0.7 (1.2)  |
> | StarCoderBase-1B        | 1.6 (1.6)  | 0.3 (1.8)  | 0.0 (1.9)  | 0.5 (1.9)  | 0.1 (2.0)  | 0.0 (2.0)  | 0.0 (2.0)  | 0.0 (1.9)  | 0.0 (1.8)  |
> | StarCoderBase-7B        | 13.9 (1.1) | 9.3 (0.7)  | 4.0 (0.5)  | 2.0 (0.5)  | 1.6 (0.2)  | 1.2 (0.2)  | 0.6 (0.3)  | 0.6 (0.1)  | 0.3 (0.2)  |
> | StarCoderBase-15.5B     | 22.9 (1.3) | 14.2 (1.3) | 11.0 (0.9) | 10.1 (0.9) | 6.1 (0.8)  | 5.8 (0.7)  | 6.8 (0.5)  | 4.5 (0.3)  | 2.6 (0.3)  |
> | DeepSeekCoder-6.7B-base | 40.3 (1.5) | 24.7 (1.2) | 20.2 (1.1) | 14.2 (1.1) | 8.5 (0.8)  | 7.2 (0.8)  | 9.6 (0.8)  | 4.6 (0.7)  | 3.2 (0.5)  |
> | GPT-4o-mini             | 79.8 (0.4) | 71.2 (0.2) | 63.6 (0.0) | 64.9 (0.2) | 52.7 (0.1) | 45.5 (0.0) | 46.1 (0.0) | 35.4 (0.0) | 27.7 (0.0) |
> | GPT-4o                  | 98.5 (1.2) | 96.7 (1.0) | 97.3 (0.6) | 97.4 (0.5) | 96.0 (0.4) | 94.9 (0.3) | 93.6 (0.2) | 91.7 (0.2) | 88.9 (0.1) |
> ```
>
>
>
> >Below Algorithm 1 is a list of the variables to be studied, n_p is not listed, should it be?
>
>
> We do list n_p in the last paragraph of section 3.3 *“n_p to ∞, 150, 50, 50 for one-step, two-step, three-step, and concatenation tasks respectively (this results in a maximum of 6000 prompts for each condition).”* The purpose of n_p is to limit the number of generated prompts when the number of combinations is huge. Please let us know if this does not address your question or if we have misunderstood.
>
>
> >In the prompt samples in the appendix it would be good to get a description in the caption of what the target and intermediate functions are
>
>
> We will revise the caption as suggested.
>
>
> >Describe the shaded region in Figure 4 in the caption
>
>
> The shaded regions represent 95% confidence intervals (same for all shaded regions in charts). We will update the caption to reflect this.
>
>
> >The colours in the colourbars for figure 6 need to be fixed
>
>
> Thank you for pointing this out—it appears to be a rendering issue in Preview and Safari (it should appear correctly in Chrome/Firefox). We will correct this in the final version.
>
>
> >Adding the names of the models in Figure 7 (perhaps on the left side) would make it more readable
>
>
> We agree—thanks for the suggestion. We will add model names to improve clarity.

---

> > ### Comment · Reviewer_dwJc · 2025-04-13
> >
> > I thank the authors for responding to my comments and acknowledging my concerns.
> >
> > Overall, the authors' proposed changes greatly clarify and improve the paper. It is reassuring to see that the GPT-4o models only very rarely returned the trivial solution. I feel my concerns have been addressed.
> >
> > Regarding the “inability to test different context windows”, yes, I meant the limitation that not all models allow for the sliding window experiment. It would be good to group that in the main limitation discussion.

---

### Review · Reviewer_bhMM · 2025-03-14

**Summary Of Contributions:**

The paper introduces a set of synthetic problems to test LLMs in a controlled way for their capabilities in handling long range dependencies in code completion scenarios. The paper evaluates 8 models (6 open-source code models and 2 GPT-4o models) on their tests and finds that the code completion models struggle with longer dependencies. The paper also analyzes the influence of context size, number of distractors, position of relevant functions, and kind of errors produced. In the second half of the paper the authors investigate how annotating the relevant functions with call graph information can improve model performance and find that this annotation helps code completion models.

**Audience:**

Yes

**Broader Impact Concerns:**

There are no ethical concerns with this paper.

**Claims And Evidence:**

Yes

**Requested Changes:**

-	I found Table 3 rather confusing: It is not clear what exactly is displayed in Table 3a (three-step, concatenation, or both). Additionally, the 51.7 should be bold instead of the 49.4. In general, both Tables took me a while to comprehend. I think a single table just displaying the performance of all comment types for both problems and templates would be easier to understand here. I also suggest providing detailed findings like in Table 11 in Appendix C for the names only runs. I see no reason not to do that.
-	A way to strengthen results would be to explore more complex structures in the calls (like control structures or loops).
-	Lastly, how many steps would be needed to notably reduce the performance of GPT-4o.

**Strengths And Weaknesses:**

-	The paper is well written and easy to understand. I only wished a few more experimental details would be explicitly stated (e.g. number of distractors and prompt length for Fig.2 a). I have a few minor suggestions that could improve readability and completeness.
-	The design of the test functions is fitting and enables for controlled experiments.
-	The paper evaluates the models in sound experiments and displays the results mostly in an easy to understand format.
-	The constructed tests are rather simple (call and concatenate). Incorporating some more complex but still easy to control structures like loops or if statements would provide further insights.


Overall, I found that the paper to be worthy of publication but the results could be further strengthened (see below).

---

> ### Author Response · Authors · 2025-03-28
>
> We thank the reviewer for their thoughtful and constructive feedback, as well as their positive assessment of the paper. We're glad that the reviewer found the paper well-written and the experimental design sound. We appreciate the detailed suggestions, which have helped us clarify the presentation and strengthen the work. We include responses to individual points below:
>
>
> >I found Table 3 rather confusing: It is not clear what exactly is displayed in Table 3a (three-step, concatenation, or both).  I think a single table just displaying the performance of all comment types for both problems and templates would be easier to understand here.
>
> Our goal with Table 3 was to separately summarize performance changes from call graph comments—first by comment template, then by task using the best-performing comment template—as that aligned with how we discussed the results in the main text. However, we agree that a unified table showing performance across all splits is clearer. We’ve implemented this change and will update Table 3 accordingly. Thank you for this helpful suggestion.
>
> Preview of updated table:
>
> ```
> +----------------+---------------+-------+-------+-----------+-------+
> | Comment Type |   Template    | none  | calls | called-by |  both |
> +--------------+---------------+-------+-------+-----------+-------+
> | Three-step   | Names Only    | 46.9  | 47.8  | 72.9      | 69.3  |
> |              | Full Sentence | 46.9  | 55.3  | 74.2      | 76.8  |
> +--------------+---------------+-------+-------+-----------+-------+
> | Concatenation| Names Only    | 23.3  | 24.6  | 29.5      | 29.5  |
> |              | Full Sentence | 23.3  | 27.6  | 32.8      | 36.4  |
> +--------------+---------------+-------+-------+-----------+-------+
> ```
>
>
>
> > I also suggest providing detailed findings like in Table 11 in Appendix C for the names only runs. I see no reason not to do that.
>
> We agree and will include a breakdown for the names-only runs in Appendix C, following the same format as Table 11.
>
>
> >A way to strengthen results would be to explore more complex structures in the calls (like control structures or loops).
>
> Thanks for the suggestion. While previous works such as [1] and [2] have investigated how models might be better equipped to handle such control structures, this has generally been in the context of shorter contexts. Understanding how that interacts with long range dependencies and the impacted coding structures would be worthwhile. We reserve these explorations for future work.
>
> [1] Exploring Length Generalization in Large Language Models, Anil et al, 2022 ([https://arxiv.org/pdf/2207.04901)](https://arxiv.org/pdf/2207.04901)
>
> [2] Show Your Work: Scratchpads for Intermediate Computation with Language Models Nye et al 2021 (https://arxiv.org/pdf/2112.00114)
>
>
>
> >Lastly, how many steps would be needed to notably reduce the performance of GPT-4o.
>
> While our primary experiments focus on function positions and order permutations (which grow factorially with step count), we conducted a follow-up experiment using a **six-step task** (double our previous maximum) to estimate the effect on GPT-4o. We sampled ~5,000 prompts per condition to maintain consistency with the main experiments. While not exhaustive enough to fully characterize all effects, these results should be indicative of the impact of the number of hops. Results are in the table below:
>
> ```
> +---------------+----------+----------+------------+----------+
> | # distractors | one-step | two-step | three-step | six-step |
> +---------------+----------+----------+------------+----------+
> |       0       |   1.00   |   1.00   |    1.00    |   1.00   |
> |       1       |   1.00   |   0.98   |    0.96    |   0.99   |
> |       5       |   1.00   |   0.95   |    0.92    |   0.60   |
> +---------------+----------+----------+------------+----------+
> ```
>
>
> As discussed in the paper, distractors are critical to revealing model limitations. In the 5-distractor setting, GPT-4o’s accuracy@3 drops from 92% (three-step) to 60% (six-step).

---

### Decision · Action_Editor_4R5b · 2025-04-22

**Recommendation:** Accept as is

**Comment:**

The paper presents a well-executed empirical investigation into how current LLMs handle long-range dependencies in code completion. The evaluation proposed by authors allow to have a strong alternative to the popular benchmark needle in the hay stack in particular for coding tasks. Having more benchmarks to be able to assess model performance on long context task is an important topic.

**Audience:**

Yes, the paper would be of clear interest to the TMLR audience. The work addresses a well-motivated issue: how large language models, particularly those for code generation, handle complex dependencies in long contexts settup.

**Claims And Evidence:**

Yes, the claims made in the paper are supported by clear and convincing experimental evidence.  All three reviewers agree that the experimental design is sound and results are well-presented.

---

> ### Author Response · Authors · 2025-05-21
>
> We sincerely thank the reviewers and the action editor for their feedback and help in improving the content of the manuscript throughout this review process and are pleased with the final outcome. We have uploaded a camera ready version that incorporates updates as discussed during the review period. This includes fixes to figures and tables as well as improving the discussion to better contextualize the work and connect it to existing literature.
>
> Thanks again to the AE for their effort in shepherding this through and to all the reviewers for their time.